# Bevacizumab, olaparib, and durvalumab in patients with relapsed ovarian cancer: a phase II clinical trial from the GINECO group

Gilles Freyer [1,2,3] ✉, Anne Floquet [2,4], Olivier Tredan[2,5], Aurore Carrot[2,6], Carole Langlois-Jacques[2,7], Jonathan Lopez[2,8], Frédéric Selle [2,9], Cyril Abdeddaim[2,10], Alexandra Leary[2,11], Coraline Dubot-Poitelon[2,12], Michel Fabbro [2,13], Laurence Gladieff[2,14] & Michele Lamuraglia [2,15]

Most patients with advanced ovarian cancer (AOC) ultimately relapse after platinum-based chemotherapy. Combining bevacizumab, olaparib, and durvalumab likely drives synergistic activity. This open-label phase 2 study (NCT04015739) aimed to assess activity and safety of this triple combination in female patients with relapsed high-grade AOC following prior platinum-based therapy. Patients were treated with olaparib (300 mg orally, twice daily), the bevacizumab biosimilar FKB238 (15 mg/kg intravenously, once-every-3-weeks), and durvalumab (1.12 g intravenously, once-every-3-weeks) in nine French centers. The primary endpoint was the non-progression rate at 3 months for platinum-resistant relapse or 6 months for platinum-sensitive relapse per RECIST 1.1 and irRECIST. Secondary endpoints were CA-125 decline with CA-125 ELIMination rate constant K (KELIM-B) per CA-125 longitudinal kinetics over 100 days, progression free survival and overall survival, tumor response, and safety. Non-progression rates were 69.8% (90%CI 55.9%-80.0%) at 3 months for platinum-resistant relapse patients (N = 41), meeting the pre-specified endpoint, and 43.8% (90%CI 29.0%-57.4%) at 6 months for platinum-sensitive relapse (N = 33), not meeting the prespecified endpoint. Median progression-free survival was 4.1 months (95%CI 3.5–5.9) and 4.9 months (95% CI 2.9–7.0) respectively. Favorable KELIM-B was associated with better survival. No toxic deaths or major safety signals were observed. Here we show that further investigation of this triple combination may be considered in AOC patients with platinum-resistant relapse.

Prognosis for patients diagnosed with advanced-stage ovarian cancer is poor. Although first-line treatment with cytoreductive surgery and platinum-based chemotherapy is often initially successful, the majority of these patients (70%) experience iterative relapses within 3 years[1,2]. The backbone of standard therapy after relapse for patients with platinum-sensitive disease is a platinum-based combination, administered for repeated lines of therapy unless platinum resistance occurs.

The anti-VEGF antibody bevacizumab is indicated in combination with standard carboplatin-based regimens in platinum-sensitive disease[3–5], and in combination with paclitaxel, pegylated liposomal doxorubicin, or topotecan in platinum-resistant patients[6].

Olaparib, an inhibitor of the enzyme poly-ADP-ribose polymerase (PARP), is indicated as maintenance therapy for ovarian cancer patients, both in the front-line setting[7,8], as well as for relapsing,

---

platinum-sensitive metastatic disease[9,10]. Synergy between olaparib and bevacizumab is hypothesized to occur through tumor environment modulation and signaling of DNA damage inhibition with the acquisition of homologous recombination deficiency (HRD) defects in a hypoxic environment leading to increased sensitivity to PARP inhibition[11]. The PAOLA-1 study confirmed a progression-free survival-(PFS) benefit with the addition of maintenance therapy with olaparib in patients with advanced ovarian cancer (AOC) receiving first-line standard therapy including bevacizumab, notably in patients with HRD-positive tumors, and irrespective of *BRCA* mutational status[8].

With responses after relapse being frequently short-lived, novel therapeutic strategies are actively being sought. Immunotherapy has dramatically altered the prognosis for several cancer indications, and tumor-infiltrating lymphocytes and expression of programmed cell death protein 1 (PD1) and its ligand PD-L1 are reported in a significant proportion of ovarian cancers[12]. Nonetheless, the success seen with single-agent anti-PD(L)−1 immunotherapy in other oncology indications has not been mirrored in ovarian cancer[13]. Combinatorial strategies have thus become the primary focus of research in this setting to leverage potential three-way synergistic interactions.

Preclinical evidence supports a potential synergistic interaction when anti-PD-(L)1 blockade is combined with PARP inhibition. Anti-PARP activity results in a bulky and toxic build-up of PARP complexes[14], releasing antigens and necrotic signals, thereby favoring an immune response. Olaparib has been reported to increase mutational load in tumor cells, which correlates with antitumor immune response[15,16]. Similarly, combining anti-PD-(L)1 blockade with anti-VEGF therapy has synergistic potential. VEGF has immunosuppressive activity in ovarian cancer[17], and anti-VEGF therapies can normalize the structure of intratumoral blood vessels, which correlates with pathologic response, and also reprograms the tumor immune microenvironment[18]. Combining these two blockade approaches, has the potential to increase the proportion of antitumor immune cells and decrease the expression of multiple immune checkpoints.

In a proof of concept study, modest clinical activity was reported with olaparib plus the anti-PD-(L)1 durvalumab in immune checkpoint inhibitor-naive patients with recurrent ovarian cancer, in a predominantly platinum-resistant population[12]. Translational analyses supported an immunostimulatory effect from the dual combination. The authors suggested that VEGF signaling may counterbalance immunostimulation, and that blocking VEGF signaling has the potential to further improve efficacy.

Here we report a phase 2 investigation of a triple combination of olaparib with durvalumab and bevacizumab in AOC in which the primary objective is to determine the non-progression rate at 3 months for platinum-resistant disease and at 6 months for platinum-sensitive disease. The study showed that combining immunotherapy with an anti-VEGF and a PARP inhibitor resulted in encouraging efficacy in patients with platinum-resistant relapse AOC.

## Results
### Patient population
A total of 74 patients were enrolled and treated between 01 March 2019 and 23 January 2020, 41 of whom had platinum-resistant relapse and 33 had platinum-sensitive relapse (Fig. 1). A median of nine cycles (range 1–32) of both bevacizumab and durvalumab were administered in the overall population, with slightly fewer median cycles of olaparib (8.5, range 1–30). Patients with platinum-sensitive disease received a higher number of cycles for all three agents compared to the platinum-resistant group. Baseline demographics, disease characteristics, and biomarkers are presented in Table 1 and Supplementary Tables 1, 2. Most patients (89%) had tumors of ovarian origin, and almost all patients (96%) had tumors with a serous histology. Germline *BRCA1/2* mutational status was available in 54 patients (73%), with eight patients (15%) having a *BRCA1* mutation and five patients (9%) having a *BRCA2*

mutation. Three patients (5%) had a somatic *BRCA1* mutation only. Seven platinum-resistant patients (17%) and ten platinum-sensitive patients (30%) had received a single line of prior systemic therapy at relapse. Platinum-resistant patients had received a median of three lines of prior therapy compared to a median of two prior lines in platinum-sensitive patients, and a higher proportion of platinum-resistant patients had received bevacizumab (85 vs 64%). On the other hand, platinum-sensitive patients were more likely to have received prior anti-PARP therapy than platinum-resistant patients (52 vs 32%, respectively).

### Radiological efficacy outcomes
Median follow-up was 15.4 months (range 1.0–21.5 months) and was similar in the platinum-resistant and platinum-sensitive groups. For the intent-to-treat (ITT) population evaluated per RECIST 1.1, in the platinum-resistant cohort, the 3-month non-progression rate was 69.8% (90% CI 55.9–80.0%). In platinum-sensitive patients, the 6-month non-progression rate was 43.8% (90% CI 29.0–57.4%) (Table 2 and Fig. 2).

Exploratory analyses of efficacy according to *BRCA1/2* status showed that in the platinum-resistant group, the 3-month non-progression rate was 72.2% (90% CI 57.8–82.4) in patients with *BRCA1/2* wild-type or missing status ($N = 37$) and 50.0% (90% CI 10.3–80.9) in *BRCA1/2* mutant ($N = 4$) patients, and the 6-month non-progression rates were 40.0% (90% CI 15.9–63.3) in *BRCA1/2* mutant patients ($N = 11$) and 45.5% (90% CI 27.7–61.6) in patients with *BRCA1/2* wild-type or missing status ($N = 22$) in the platinum-sensitive group (Table 2). Ad hoc analyses were performed according to prior PARP inhibitor exposure; in the platinum-sensitive cohort, median PFS was 4.2 months in patients with prior PARP inhibitor exposure ($N = 17$) compared with 6.7 months in those without ($N = 16$), and in the platinum-resistant cohort median PFS was 4.2 months versus 4.1 months ($N = 28$ vs $N = 17$), respectively. Response data suggested that patients in the platinum-sensitive cohort with prior exposure were less likely to respond than those without (Supplementary Table 3). A swimmer plot showing time on treatment according to platinum-resistant and platinum-sensitive disease, response, prior PARP inhibitor, and *BRCA* status is shown in Fig. 3. Eight patients in each cohort were still on treatment at the cutoff date.

Median PFS (per RECIST 1.1 and/or clinical progression) was 4.1 months (95% CI 3.5–5.9) in platinum-resistant patients and 4.9 months (95% CI 2.9–7.0) in platinum-sensitive patients. Exploratory analyses per *BRCA* mutation status showed no significant difference ($P = 0.79$) in PFS when comparing patients with a *BRCA* mutation ($N = 15$) versus those without a known *BRCA* mutation ($N = 59$) in the overall population (Supplementary Fig. 1). Efficacy outcomes were similar using irRECIST, with higher non-progression rates and longer median PFS in both groups (Table 3). Median overall survival (OS) was 18.8 months (95% CI 9.6-not reached) and 18.5 months (95% CI 15.6-not reached), respectively.

### Predictive value of KELIM-B CA-125 decline
The potential predictive value of CA-125 ELIMination rate constant K (KELIM-B) was analyzed in terms of OS and PFS. Median CA-125 decline was 281 kU/L (range, 10–25,000) in platinum-resistant patients and 43 kU/L (range 3–12,000) in platinum-sensitive patients. Of the 74 patients treated, 62 (84%) were eligible for KELIM-B estimation (35 platinum-resistant; 27 platinum-sensitive) with a total of 247 CA-125 measurements (median of three CA-125 titers per patient); all were assessed for OS, and 44 were assessed for PFS (i.e., 18 progressed within 100 days). KELIM-B demonstrated prognostic value with a median PFS of 1.5 months (95% CI 0.7–6.2) for unfavorable KELIM-B versus 6.2 months (95% CI 3.5–9.0) for favorable KELIM-B ($P = 0.03$; HR 0.47, 95% CI 0.24–0.92) (Fig. 4a). Similarly, median OS was 10.6 months (95% CI 6.3-not reached) for unfavorable KELIM-B and was not reached in the favorable group ($P = 0.003$; HR 0.29, 95% CI 0.12–0.69) (Fig. 4b).

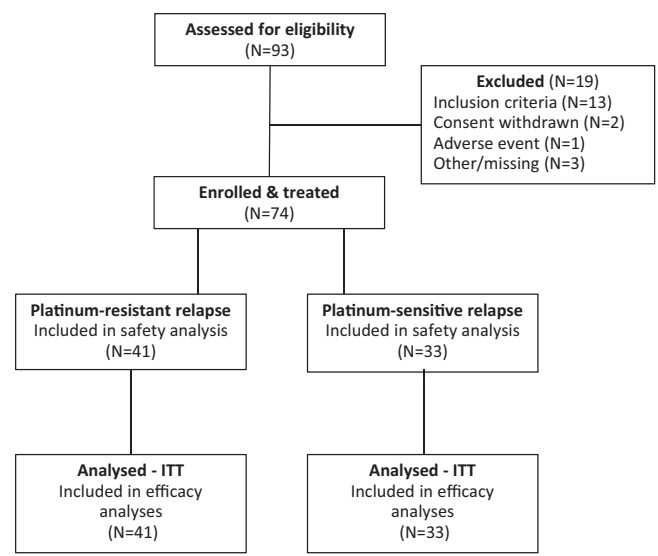

**Fig. 1 | Flow-chart of the BOLD study.** Description of the patients screened for the study.

## Table 1 | Baseline patient and disease characteristics, overall and by platinum status

| | Platinum-resistant relapse N = 41 | Platinum-sensitive relapse N = 33 | Total N = 74 |
|---|---|---|---|
| Age (years), median (range) | 66 (38–89) | 65 (43–81) | 65.5 (38–89) |
| ECOG performance status, N (%) | | | |
| 0 | 20 (49%) | 24 (73%) | 44 (59%) |
| 1 | 21 (51%) | 9 (27%) | 30 (41%) |
| Tumor origin, N (%) | | | |
| Ovarian | 37 (90%) | 29 (88%) | 66 (89%) |
| Primary peritoneal | 4 (10%) | 4 (12%) | 8 (11%) |
| Histology, N (%) | | | |
| High-grade serous | 38 (93%) | 33 (100%) | 71 (96%) |
| Other* | 3 (7%) | 0 | 3 (4%) |
| BRCA1/2 mutation | 4 (10%) | 11 (33%) | 15 (20%) |
| Germline BRCA mutation | N = 27 | N = 27 | N = 54 |
| BRCA1 | 2 (7%) | 6 (22%) | 8 (15%) |
| BRCA2 | 2 (8%)** | 3 (11%) | 5 (9%) |
| Somatic BRCA mutation (isolated) | N = 28 | N = 29 | N = 57 |
| BRCA1 | 0 | 3 (10%) | 3 (5%) |
| BRCA2 | 0 | 0 | 0 |
| Prior systemic therapy, N (%) | | | |
| N lines chemotherapy, median (range) | 3 (1–8) | 2 (1–8) | 2 (1–8) |
| Antiangiogenic agent | 36 (88%) | 28 (85%) | 64 (86%) |
| Bevacizumab | 35 (85%) | 21 (64%) | 56 (76%) |
| Other anti-angiogenic agent | 11 (27%) | 8 (24%) | 19 (26%) |
| PARP inhibitor | 13 (32%) | 17 (52%) | 30 (41%) |
| Olaparib | 4 (10%) | 9 (27%) | 13 (18%) |
| Niraparib | 7 (17%) | 8 (24%) | 15 (20%) |
| Rucaparib | 2 (5%) | 0 | 2 (3%) |
| Platinum-free interval (months), mean (SD) | 4.2 (1.8) | 8.5 (1.7) | NA |

PARP poly-ADP ribose polymerase, NA not applicable.
*Undifferentiated, endometrioid, other.
**Missing data for 1 patient.

## Exploratory analyses of immune-related biomarkers

A range of biomarkers were analyzed for predictive value in terms of outcome, with tumor inflammation signature (TIS) suggesting potential for predictive value (see Supplementary Fig. 2). Median TIS based on gene expression profiles using RNA from baseline tumor samples was available for 53 patients (27 platinum-resistant, 26 platinum-sensitive), with median values of 6.5 (range 4.4–8.8) and 6.2 (range 3.9–8.9), respectively (Supplementary Table 1). In each cohort, patients were dichotomized based on the median cohort value. Data suggest that inflammatory biomarkers may predict for PFS and OS in the overall population, with a higher TIS (i.e., more inflamed) associated with better survival (median PFS 5.9 months, 95% CI 3.5–9.5 vs 4.0 months, 95% CI 1.5–4.2; median OS not reached, 95% CI 11.9–not reached vs 10.4, 95% CI 6.7–not reached) (Fig. 4c, d). This appeared more pronounced for the platinum-sensitive group (median PFS 6.9 months, 95% CI 3.3–10.3 vs 4.0 months, 95% CI 1.4–5.7) (Fig. 4c). Analysis of TIS according to response also suggests a potential relationship between a higher TIS in responding patients in the platinum-sensitive group (Supplementary Fig. 3). A combination of KELIM-B and TIS carried stronger predictive value than either parameter alone, and was most pronounced when comparing both parameters as favorable or both as unfavorable, for PFS as well as for OS (Fig. 4e, f). See also Supplementary Data 1, 2.

## Safety

All 74 treated patients experienced at least one adverse event (AE), with the majority of patients experiencing at least one related AE (38 patients [93%] in the resistant cohort; 32 patients [97%] in the sensitive cohort). The most common AEs (≥20%) irrespective of causality were asthenia (80%), nausea (65%), anemia and abdominal pain (46% each), diarrhea (45%), decreased appetite (39%), arthralgia (35%), and dyspnea (34%). Grade ≥3 AEs were reported in 19 patients (26%), notably anemia (19%), hypertension (12%), asthenia and general health deterioration (8%), dyspnea, and pulmonary embolism (5%)(Table 4). Four patients experienced Grade 4 AEs, including lipase increase (two events), transaminase increased, stroke, and neutropenia. Of note, there were no reports of pneumonitis, interstitial lung disease, leukemia, or myelodysplasia. Hypothyroidism and hyperthyroidism were reported in 15 patients (20%; including one grade 3 case) and 8 (11%) patients, respectively, and a single case of thyroiditis (grade 1) was reported. Transaminase increases were reported in three patients (one each of grades 2, 3, and 4). Skin toxicity was reported in 19 patients (26%), all cases of which were grade 1–2. Seventeen patients (23%) had hypertension, nine (12%) of whom had grade 3. Seven patients (10%) had proteinuria (one grade 3) and five patients reported pulmonary embolism (one grade 2, four grade 3). Infusion-related reaction, anaphylactic reaction, and drug hypersensitivity were infrequent (one patient each, all grade 1–2). Grade 3 bacterial colitis and grade 3 pyelonephritis were reported in one patient each, and another had grade 3 hemophagocytic lymphohistiocytosis leading to treatment discontinuation, but recovered with medication. No treatment-related deaths occurred. Twelve patients (16%) withdrew from treatment due to an AE, including anemia (six patients – olaparib), bowel obstruction (two patients), transaminase increase, ischemic stroke, neutropenia, and pulmonary embolism (one patient each); all three drugs were withdrawn other than for anemia (olaparib only) and neutropenia (olaparib only).

**Table 2 | Efficacy (RECIST 1.1 and clinical progression) per investigator assessment in ITT patients, by platinum status**

| | Platinum-resistant relapse N = 41 | Platinum-sensitive relapse N = 33 |
|---|---|---|
| Non-progression rate (%), median [90% CI] | 3 months | 6 months |
| All patients | 69.8 [55.9–80.0] | 43.8 [29.0–57.4] |
| BRCA1/2mut | N = 4 | N = 11 |
| | 50.0 [10.3–80.9] | 40.0 [15.9–63.3] |
| BRCA1/2wt / missing status | N = 37 | N = 22 |
| | 72.2 [57.8–82.4] | 45.5 [27.7–61.6] |
| Best overall response (confirmed)* | | |
| Complete response | 0 | 1 (3%) |
| Partial response | 11 (28%) | 11 (34%) |
| Stable disease | 18 (46%) | 16 (50%) |
| Progressive disease | 10 (26%) | 4 (13%) |
| Not evaluable | 2 | 1 |
| Objective response rate (%) [95% CI]* | 11 (28%) [15–45] | 12 (38%) [21–56] |
| Median PFS, in months [95% CI] | 4.1 [3.5–5.9] | 4.9 [2.9–7.0] |
| Median OS, in months [95% CI] | 18.8 [9.6 - NR] | 18.5 [15.6 - NR] |

NR not reached
*In evaluable patients

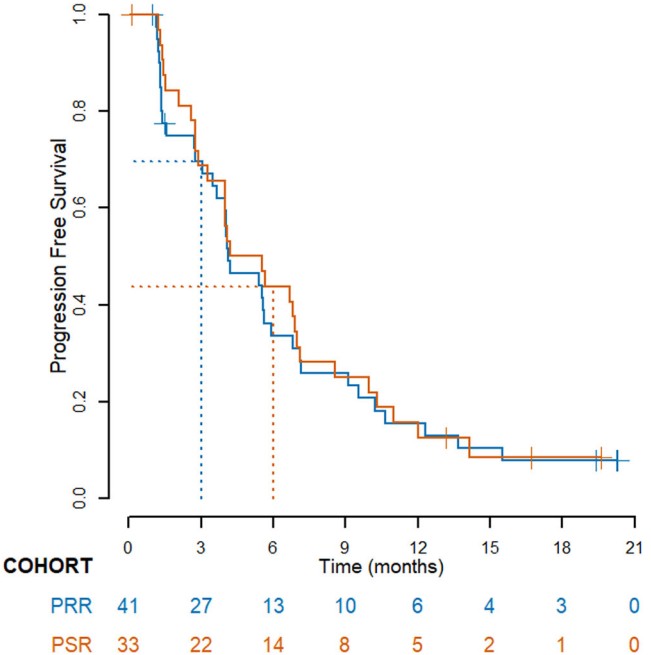

**Fig. 2 | Kaplan–Meier estimates of PFS in the platinum-resistant relapse (PRR) and platinum-sensitive relapse (PSR) cohorts.** Number of patients at risk is shown. Source data are provided as a Source Data file.

## Discussion

The use of PARP inhibitors and bevacizumab is a routine part of the therapeutic management of AOC, with bevacizumab and olaparib both indicated in relapsed AOC. Current attempts to address the lack of efficacy of immune checkpoints inhibitors in ovarian cancer include a focus on combinatorial strategies. Combining immunotherapy in a triple therapy with a PARP inhibitor and an antiangiogenic agent is one approach to address the dismal outcome for women with AOC. In our cohort of AOC patients with heavily pretreated platinum-resistant relapse, two-thirds of the population achieved durable benefit with a non-progression rate at 3 months of 69.8% (90% CI 55.9–80.0%) with the triple combination of durvalumab plus olaparib and bevacizumab. The null hypothesis that a 50% rate of non-progressive disease at 3 months was undesirable compared with historical controls in this patient population, was rejected. The overall disease control rate of 74% was consistent with that reported in a similar population in a recent pilot study of olaparib and durvalumab, including primarily platinum-resistant and heavily pretreated patients[12]. Meaningful durable clinical benefit was seen with the triple combination in our heavily pretreated platinum-resistant relapse population, with a median of three prior lines of chemotherapy, the majority (85%) of whom were exposed to prior bevacizumab and approximately one-third to a prior PARP inhibitor. The addition of durvalumab to olaparib and bevacizumab in our study population also compares favorably with the addition of different checkpoint inhibitors to current treatment options in the platinum-resistant AOC setting. Lee et al. recently reported data for a randomized trial in platinum-resistant patients, comparing the cediranib plus olaparib and durvalumab triplet with chemotherapy. The study was negative and the PFS was 2.9 months with the triplet, confirming that cediranib is unlikely to be a valuable antiangiogenic agent in ovarian cancer[19]. Avelumab administered with or without chemotherapy proved disappointing in the JAVELIN 200 study in platinum-resistant relapse patients, showing ORRs of 8 and 4% respectively[20]. The addition of pembrolizumab to niraparib in the phase 1/2 TOPACIO study gave an ORR of 18%[21], and adding nivolumab to bevacizumab monotherapy in

the same setting did not improve outcome[22]. Several practice-changing studies on this population have been published. In the AURELIA trial, patients with first or second platinum-resistant relapse received chemotherapy alone or combined with bevacizumab. Median PFS with bevacizumab was 6.7 months and median OS was 16.6 months[6]. In the phase 2 SORAYA trial, all 106 patients enrolled had received prior bevacizumab, 51% had three prior lines of therapy, and 48% received a prior PARP inhibitor. Median PFS and OS with mirvetuximab soravtansine were 4.3 months and 13.8 months, respectively[23]. In the randomized phase 3 MIRASOL trial in 453 platinum-resistant patients with high FRα expression and up to three prior lines, median PFS and OS were longer (5.6 and 16.5 months, respectively) with mirvetuximab soravtansine compared to investigator's choice[24]. With PFS and OS of 4.1 and 18.8 months, respectively, in our platinum-resistant population, half of whom had more than three prior lines and one-quarter with prolonged response duration, our results compare well with these studies and merit further evaluation. The main challenge remains to better define the patient subgroup who would obtain the greatest benefit from the triplet.

Among our population with platinum-sensitive relapse, the non-progression rate at 6 months was 43.8% (90% CI 29.0–57.4%). Unlike the platinum-resistant population, this outcome did not exclude the null hypothesis (<65% rate of non-progression). The disappointing result observed in the platinum-sensitive cohort may have been influenced by the lower-than-planned number of patients included in this cohort. However, simulations showed that the outcome would not have been substantially modified by including 40 patients rather than 33 initially planned. The 37% response rate in this population, was low compared to preliminary data for the phase 2 MEDIOLA study with the triple combination of olaparib and durvalumab plus bevacizumab[25]. In the 16 patients who were not previously exposed to PARP inhibitors in our cohort, the ORR was 44% and median PFS was 6.7 months, compared to the MEDIOLA cohort where the confirmed ORR was 77% and median PFS was 14.7 months, 95% CI 10.0–18.1; however, results from our study should be interpreted with caution given the post-hoc nature of our analysis and the small sample sizes. This may be due to the inclusion in our study of patients without measurable disease (i.e., with pleural effusion), potentially reflecting a population with a poorer

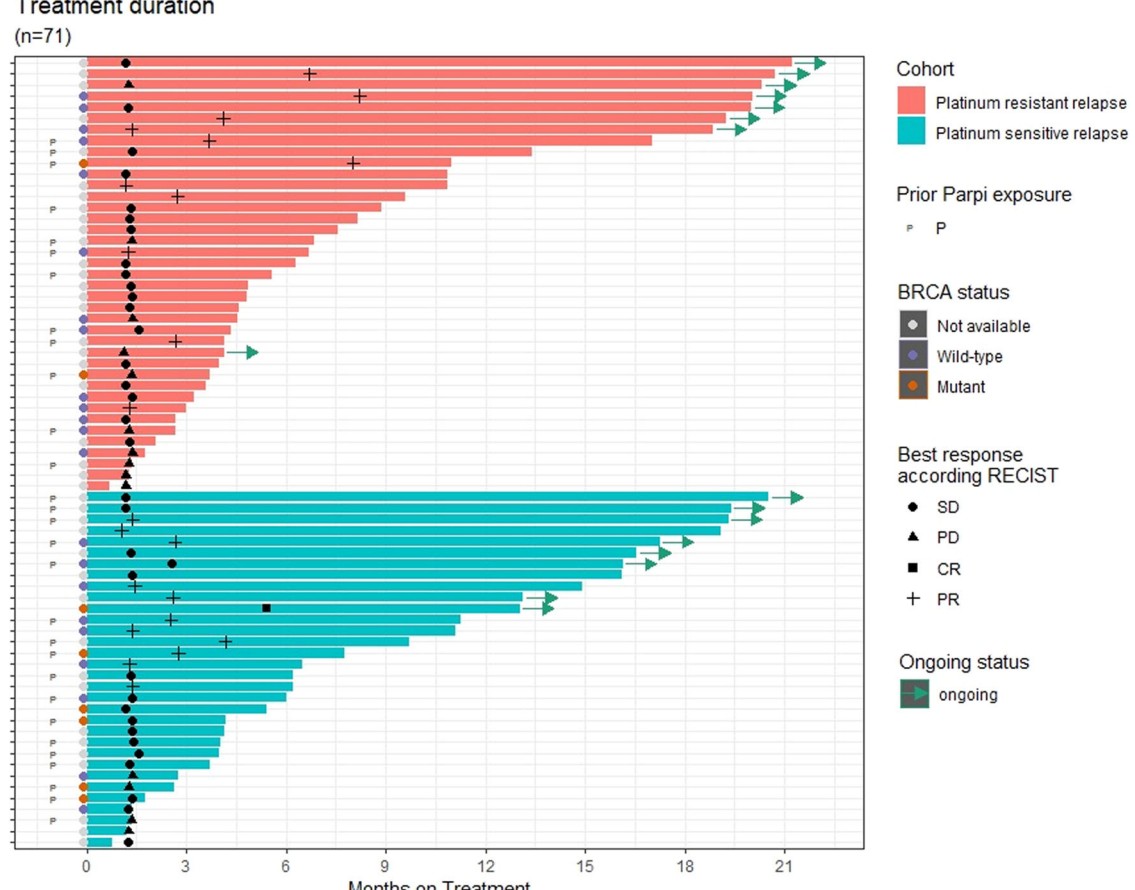

**Fig. 3 | Time on treatment, duration of response, and best overall response per RECIST 1.1, by patient.** Each horizontal bar represents a treated patient and arrows indicate treatment was ongoing at the data cutoff date. P indicates patients who received prior PARP inhibitor therapy. Three patients were not evaluable for response per RECIST 1.1 as progression was clinically symptomatic. Note: for two patients who experienced disease progression, treatment was maintained in the context of clinical benefit per protocol, given that further progression was not observed in subsequent evaluations and clinical status remained stable. Source data are provided as a Source Data file.

**Table 3 | Efficacy (irRECIST) per investigator assessment in ITT patients, by platinum sensitivity status**

|  | Platinum-resistant relapse N = 41 | Platinum-sensitive relapse N = 33 |
|---|---|---|
| Non-progression rate (%), median [90% CI] | 3 months | 6 months |
| All patients | 77.5 [64.3-86.3] | 56.1 [40.5-69.1] |
| BRCA1/2mut | N = 4 | N = 11 |
|  | 50.0 [10.3–80.9] | 50.0 [23.0-72.1] |
| BRCA1/2wt/missing status | N = 37 | N = 22 |
|  | 80.6 [66.8-89.0] | 58.7 [39.5-73.7] |
| Best overall response (confirmed)* |  |  |
| Complete response | 1 (2%) | 2 (6%) |
| Partial response | 9 (22%) | 12 (36%) |
| Stable disease | 24 (59%) | 14 (42%) |
| Progressive disease | 6 (15%) | 4 (12%) |
| Not evaluable | 1 | 1 |
| Objective response rate (%) [95% CI]* | 10 (25%) [13–41] | 14 (44%) [26–62] |
| Median PFS, in months [95% CI] | 5.4 [4.0–7.2] | 7.0 [3.3 NR] |

NR not reached.
* In evaluable patients.

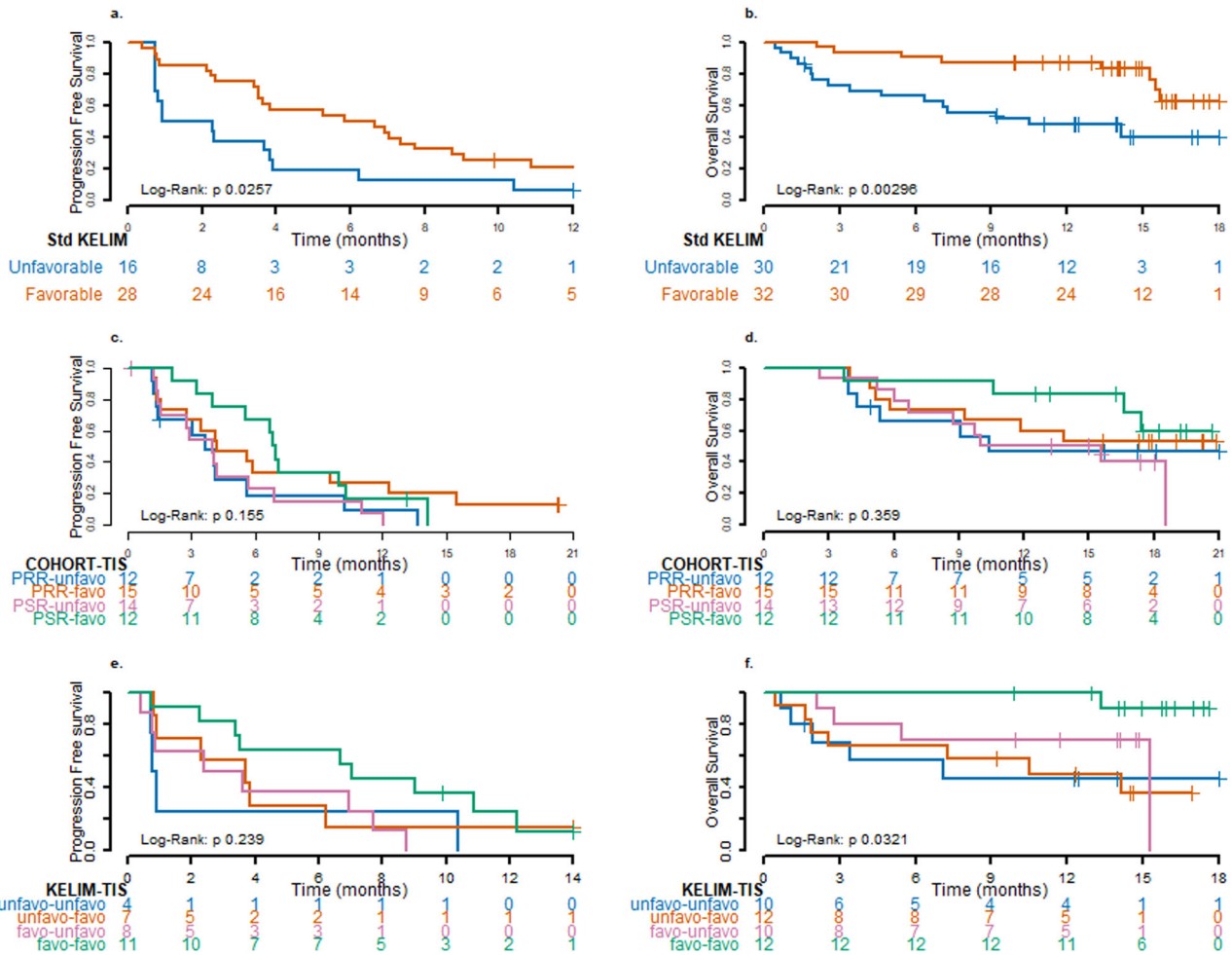

**Fig. 4 | Kaplan–Meier estimates for predictive biomarkers.** PFS (**a**) and OS (**b**) according to favorable versus unfavorable standardized (std) KELIM-B (≥1 or <1, respectively). PFS (**c**) and OS (**d**) according to favorable and unfavorable TIS (≥median and <median, respectively) in the platinum-resistant relapse (PRR) versus platinum-sensitive relapse (PSR) cohorts. PFS (**e**) and OS (**f**) according to combined favorable and unfavorable standardized KELIM-B and TIS in evaluable patients. Number of patients at risk is shown. Log-rank two-sided test: **a** Chisq = 4.979 on 1 ddl, **b** Chisq = 8.833 on 1 ddl, **c** Chisq = 5.243 on 1 ddl, **d** Chisq = 3.221 on 1 ddl, **e** Chisq = 4.215 on 1 ddl, **f** Chisq = 8.797 on 1 ddl. Source data are provided as a Source Data file.

prognosis. Of note, in the platinum-sensitive population treated with olaparib maintenance therapy in the OreO/ENGOT Ov-38 study, preliminary data reported median PFS ranging from 4.3 to 5.3 months[26]. Our results should be considered in light of recent data reported by Kim et al. with a triple combination of olaparib, pembrolizumab, and bevacizumab in patients with platinum-sensitive disease. In a small population of 44 patients, they obtained a median PFS of 22.4 months, which is encouraging[27]. Exploratory analyses in our study showed a similar level of activity with the triplet combination in platinum-sensitive patients who had received prior PARP inhibitor therapy, with a median PFS of 4.2 months in the 17 patients with prior PARP inhibitor exposure. This may be linked to synergy between the three drugs, or to dominant efficacy driven by one or two of the agents. In the OReO/ENGOT Ov-38 study, PARP inhibitor therapy showed efficacy after progression under this treatment[26]. While non-progression rates were better in the *BRCA* wild-type population for both platinum-sensitive and resistant patients in our study, interpretation of these results is again limited by the very small sample size, with both populations being predominantly *BRCA* wild-type (90% in the resistant population and 67% in the sensitive population).

Clinically, the triplet combination was well tolerated in our population of heavily pretreated relapsing AOC patients. The main toxicities were asthenia/fatigue, anemia, gastrointestinal, and dyspnea, with the main severe toxicities being anemia and hypertension. There were no signs of cumulative toxicity from the triple combination, and no new safety signals were identified. This safety profile seen with the triplet is similar to that reported with both the doublet and the triplet therapy in the MEDIOLA study[25,28]. The low rate of immunotoxicities including pulmonary interstitial disease, colitis, dysthyroidism, myocarditis, nephritis, immune skin reaction, and hypophysitis was notable, while skin toxicity was mild to moderate.

The identification of predictive biomarkers is an important aspect of current drug development programs. KELIM™ estimation is a recently developed tool which may be exploited in the ovarian cancer setting to identify patients who are more likely to benefit from treatment, notably with respect to PFS and OS with bevacizumab maintenance treatment in the first-line setting[29,30]. Our results support the usefulness of this tool in the relapsed setting, with KELIM-B predicting for patients more likely to achieve benefit with the triple therapy, and demonstrates the predictive value of KELIM-B in a chemotherapy-free regimen. Use of KELIM™ may therefore be exploited to identify those patients least likely to benefit from the planned therapy. We also identified a potential positive association between elevated T cell inflamed gene expression profile from the NanoString panel and

**Table 4 | Main adverse events with treatment with bevacizumab, olaparib, and durvalumab occurring in ≥20% of patients**

| Preferred Term | N patients (N = 74) | |
|---|---|---|
| | All grades | Grade 3–4 |
| Any AE | 74 (100%) | 19 (26%) |
| Asthenia | 59 (80%) | 6 (8%) |
| Nausea | 48 (65%) | 1 (1%) |
| Anemia | 34 (46%) | 14 (19%) |
| Abdominal pain | 34 (46%) | 3 (4%) |
| Diarrhea | 33 (45%) | 1 (1%) |
| Decreased appetite | 29 (39%) | 1 (1%) |
| Arthralgia | 26 (35%) | 0 |
| Dyspnea | 25 (34%) | 4 (5%) |
| Constipation | 24 (32%) | 0 |
| Vomiting | 24 (32%) | 1 (1%) |
| Headache | 18 (24%) | 0 |
| Cough | 17 (23%) | 0 |
| Hypertension | 17 (23%) | 9 (12%) |
| Neutropenia | 15 (20%) | 3 (4%) |
| Hypothyroidism | 15 (20%) | 1 (1%) |

improved survival outcomes, notably for the platinum-sensitive population. The role of this biomarker in ovarian cancer remains unclear with differing methodologies and cancer populations, with limited published data on ovarian cancer and conflicting reports in the literature of its predictive value[31–33]. Nonetheless, these exploratory data appear promising and this association merits further exploration, including prospective biomarker collection and analyses.

Limitations of our study include its single-arm nature, and a minority of patients with *BRCA* mutant status in both populations, limiting the interpretation of the results. Baseline data for *BRCA* status were not systematically available, and the study was not powered to show statistical differences in these subgroups nor for other biomarkers. Combined with the small number of patients, the interpretation of any association between outcomes and *BRCA* status or other translational endpoints remain hypothesis-generating. It should be noted that no evidence of the correlation between *BRCA* status and efficacy was seen with olaparib administered as re-challenge after responding to platinum-based chemotherapy following maintenance olaparib, possibly due to acquired resistance mechanisms to PARP inhibition, such as reversion mutations[26]. The small sample size limits interpretation in the subpopulation of patients with prior PARP inhibitor exposure, an increasingly prominent population given their use in the upfront maintenance setting. The use of archival tissue for translational research may have resulted in analyses that did not reflect the true tumor biology status at the time of treatment in our study, potentially introducing bias in translational interpretations. Finally, the assessment of tumor response by irRECIST was originally designed to capture delayed or flare-type responses to immunotherapy that tend to be overlooked with conventional methodology such as RECIST[34]. Nonetheless, applying irRECIST in the clinic has proven challenging; practices have changed and the irRECIST classification is not recommended in any guidelines for ovarian cancer patients treated with immunotherapy. Published data for phase 3 trials in ovarian cancer (JAVELIN 100 and 200, ATALANTE, IMAGYN 050) report response per RECIST 1.1[20,35,36], and furthermore, PFS determined by irRECIST has not shown improved predictive value for OS compared with PFS by RECIST 1.1[34].

Studies evaluating different triplet combinations are underway[37]. Immunotherapy with or without chemotherapy has given disappointing results in both chemotherapy-naive and relapsed AOC, showing no

improvement in terms of survival outcomes[20,35]. These disappointing results nonetheless highlight the importance of biomarkers to select patients for future studies with immunotherapy-based combinations in both platinum-resistant and sensitive AOC settings. Novel and promising combination approaches include the addition of the antibody-drug conjugate mirvetuximab soravtansine to bevacizumab giving high and durable responses in patients with FRα-positive AOC[38], and the Wee1 inhibitor adavosertib administered alone or in combination with olaparib, which demonstrated efficacy in patients resistant to PARP inhibitors[39]. Other agents, including XMT-1536 (a NAPi2B inhibitor), relacorilant, and alpelisib, are under investigation.

In conclusion, our study combining immunotherapy with an anti-VEGF and a PARP inhibitor supports an encouraging degree of efficacy in patients with platinum-resistant relapse AOC, whereas results were disappointing in our heavily pretreated patients with platinum-sensitive relapse. The triple combination of olaparib plus bevacizumab and durvalumab was well tolerated in relapsed AOC patients, without excess toxicity due to the combination. Further evaluation in this setting after first-line chemotherapy and maintenance with a PARP inhibitor, with or without bevacizumab, is warranted. Such studies would benefit from biomarkers to improve patient selection to optimize the success of this triple combination in this challenging population.

## Methods

### Study design

The GINECO BOLD study was an open-label, parallel cohort, single-arm phase 2 study conducted in nine French centers. The study design and conduct complied with all relevant regulations regarding the use of human study participants, and was conducted in accordance with Good Clinical Practice and the criteria set by the Declaration of Helsinki. The protocol was approved by the French ethics committee "Comité de Protection des Personnes Sud-Est I". All patients gave written informed consent before inclusion. The trial was preregistered in the EudraCT database under the number 2018-002281-39 on 19 July 2018, and submitted to the clinicaltrial.gov registry under number NCT04015739 on 02 January 2019 (posted 11 July 2019 after meeting the QC criteria). See Supplementary Note 1 (in the Supplementary Information) for the full protocol.

### Patients

The first patient was enrolled on 01 March 2019 and the last patient on 23 January 2020. Female patients aged ≥18 years with histologically confirmed, relapsed ovarian, primary peritoneal, and/or fallopian-tube high-grade carcinoma, not amenable to cytoreductive surgery, were eligible. Platinum-resistant relapse was defined as disease progression <6 months after the last platinum dose and ≥1 line of previous platinum and taxane-containing chemotherapy. Platinum-sensitive relapse was defined as disease progression ≥6 months after the last platinum dose in any prior line. Other inclusion criteria included Eastern Cooperative Oncology Group performance status 0 or 1, adequate hematologic, renal, and hepatic function, and blood coagulation parameters, a recent (<3 months) biopsy post-last chemotherapy, and ≥1 measurable or evaluable lesion. Key exclusion criteria were immunosuppressive medication within 14 days of treatment initiation, prior treatment with anti-PD(L)−1 immunotherapy (including durvalumab) active or prior autoimmune or inflammatory disorders, allogenic bone marrow transplant, history of myelodysplastic syndrome/acute myeloid leukemia, of interstitial lung disease or significant lung or cardiovascular disease. Patients could have previously received either bevacizumab or olaparib but not a combination of them.

### Treatment

Olaparib 300 mg was administered orally twice daily. FKB238 (bevacizumab biosimilar; Centus Biotherapeutics, Cambridge, UK) 15 mg/kg was administered once every 3 weeks (Q3W) intravenously (initially

90 min, subsequently 60, then 30 min if well tolerated). Durvalumab 1.12 g was administered Q3W, 1-h intravenous infusion, more than 1 h after olaparib, starting from Cycle 1. Subsequent infusion durations could be reduced. Up to two olaparib dose reductions (to 250 mg, then 200 mg) were permitted for toxicity. No dose reductions were permitted for durvalumab or bevacizumab. Treatment was administered in 21-day cycles until progression, unacceptable toxicity, or withdrawal of consent, for up to 2 years.

## Clinical assessments

AEs and clinical laboratory tests were evaluated throughout treatment per the National Cancer Institute Common Terminology Criteria for Adverse Events (NCI-CTCAE), v5.0. Tumor response was evaluated by radiological imaging per RECIST 1.1 and irRECIST, at baseline and every 6 weeks until progression, per the investigator. Clinical progression was defined as symptoms considered by the investigator as disease-related. Survival status was followed up every 3 months for up to 1 year. Serum CA-125 levels were determined every 6 weeks.

## KELIM-B modeling

The modeled CA-125 ELIMination rate constant K (KELIM-B), calculated with CA-125 longitudinal kinetics during the first 100 days of therapy, is a validated early marker of tumor chemosensitivity. The mathematical modeling of early CA-125 kinetics with a non-linear mixed effect model and KELIM™ estimation has been previously described in refs. 29,30,40. At least three CA-125 values during the first 100 days of treatment were required to ensure an accurate assessment of KELIM™ with the kinetic-pharmacodynamic model[41]. To normalize the distribution of CA-125 concentrations, and eliminate right-skewness in this distribution, CA-125 levels were log-transformed. KELIM-B was assessed as a discrete covariate standardizing KELIM-B by the platinum-sensitive and resistant median KELIM-B then separating patient populations by 1 as favorable ($\geq$1) or unfavorable (<1). See Supplementary Methods for a description of the semi-mechanistic kinetic-pharmacodynamic (k-pd) model adjustment and qualification.

## Biomarker analyses

The tumor inflammation signature (TIS) (see Supplementary Methods) was evaluated with the Nanostring IO360 immuno-oncology panel (770 genes) using RNA from formalin-fixed paraffin-embedded baseline tumor samples (archival or fresh obtained <3 months prior to treatment start and after prior chemotherapy) based on 18 genes, as described in ref. 42. In the absence of an established cutoff, patients were dichotomized as favorable on the basis of the median TIS (TIS $\geq$ median; i.e., higher inflammation) or unfavorable (TIS < median; i.e., lower inflammation).

## Outcomes

The primary objective was to determine the efficacy of the triple combination in patients with relapsed high-grade, ovarian, fallopian tube, or peritoneal cancer, measured as the rate of radiological (according to RECIST 1.1) and clinical non-progressive disease at 3 months in platinum-resistant patients and at 6 months in platinum-sensitive patients, per investigator. Secondary objectives were to determine CA-125 decline (according to KELIM-B), PFS, OS, tumor response, and safety in these populations. Translational objectives included the correlation of immune-related biomarkers with efficacy outcomes. The protocol planned to assess tumor response and tumor progression according to both RECIST and irRECIST for the primary objective. However, irRECIST was subsequently considered less appropriate for reporting the final results due to a lower level of evidence[34]. We therefore chose to present more detailed data using RECIST 1.1.

## Sample size calculation, statistical methods, and reproducibility

A one-stage design and the exact binomial distribution was used, and the sample size was calculated independently in the two cohorts[43]. For the platinum-resistant cohort, the objective was to exclude a 3-month non-progressive disease rate of $\leq$50%, with a positive hypothesis of 75%. A total of 23 evaluable patients yields a maximum one-sided type-1 error rate of $\alpha$ = 5% and a power of $\geq$80% when the true non-progressive disease rate is 75%. For the platinum-sensitive cohort, the objective was to exclude a 6-month non-progressive disease rate of $\leq$65%, with a positive hypothesis of 84%. A total of 40 evaluable patients yields a one-sided type-1 error rate of $\alpha$ = 3% maximum and a power of $\geq$82% when the true non-progressive disease rate is 84%.

The discrepancy between the planned and actual numbers of patients arose during accrual, following routine data monitoring when 23 platinum-resistant patients and 33 platinum-sensitive were registered in the database as planned in the protocol. Monitoring identified that five patients were incorrectly registered in the database as platinum-sensitive, whereas they were actually platinum-resistant, and the true number of patients at that time, was, in fact, 28 platinum-resistant (i.e., 23 + 5), and 28 in the platinum-sensitive cohort. Given the increasing success of the protocol, another 18 patients had already been screened (13 platinum-resistant and 5 platinum-sensitive), and were included and treated on the Steering Committee's recommendation. However, no further study drug was available for any additional patients, and the study was closed. Therefore, the final numbers of patients included were 41 platinum-resistant (i.e., 28 + 13), and 33 platinum-sensitive (i.e., 28 + 5) patients. Patients were analyzed according to their actual prior platinum status, given that the treatment was identical in the two cohorts, along with the non-randomized parallel cohort single-arm study design. No data were excluded from the analyses.

For non-progression rates, patients who died due to a cause other than disease progression were censored at the time of death. Patients without progression who started an alternative therapy were censored at that time. Patients lost to follow-up or alive without progression or a new therapy were censored at the date of the last follow-up. The Kaplan–Meier method was used to estimate time-related parameters. KELIM-B survival analyses (Kaplan–Meier and log-rank tests) were implemented with a landmark time point set at 100 days after treatment started to avoid biases related to links between early progression and CA-125 kinetics, using a two-sided 0.05 alpha risk. The predictive value of Std KELIM-B and/or TIS/std KELIM-B for PFS and OS were assessed using Cox regression and log-rank or Mann–Whitney tests. Efficacy was analyzed in the ITT population (all included patients regardless of whether they received treatment), and safety in all treated patients. Descriptive analysis were performed with SAS v9.4. R software v4.1.1 (packages survival, ggplot2, swimplot) was used to perform survival analyses, the curves and the swimmer plot, with a data cutoff date of 15 February 2021. Confidence intervals (CI) were calculated using the Clopper-Pearson method. For KELIM-B analyses, NONMEM 7.5.0 software (ICON Development Solutions) was used to fit the semi-mechanistic model to CA-125 kinetic data, the SAEM algorithm with Monte Carlo IMP (importance sampling) for standard errors, and R software v4.1.1 for statistical analyses. Data were collected using Ennov Clinical version 7.5.730.

## Reporting summary

Further information on research design is available in the Nature Portfolio Reporting Summary linked to this article.

# Data availability

Data sharing in a public repository was not planned at the start of the study. Per European and French regulations for personal data privacy, this is not permitted without having informed the study

participants which was not done. This is also linked to a confidentiality agreement with AstraZeneca who provided the drug and funding. This agreement aims to guarantee protection for the company about potential sub-licensable or patentable information/discovery. Requests to access the deidentified data for further scientific use can be sent to ARCAGY-GINECO (Sébastien Armanet sarmanet@arcagy.org) and will be considered on a case-by-case basis in a timely manner beginning 3 months and ending 5 years after this article publication. The request must contain a proposal with scientific and methodologically justified objectives. A Data Transfer Agreement will be established to provide a formal framework regarding the use of the data. The deidentified data underlying the results generated in this article are provided in the Source Data, Supplementary Data 1, 2. The study protocol and statistical analysis plan are available in the Supplementary Information. The remaining data are available within the Article, Supplementary Information, or Source Data file. Source data are provided as a Source Data file. Source data are provided with this paper.

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

## Acknowledgements

We thank all the patients who participated in the trial and their families. We acknowledge Mihary Andriamamonjy, Sébastien Armanet, Maxime Bonjour, Erika Cantelli, Awa Cisse-Ba, Ariane Guilbard, Laure Jerber, Christine Montoto-Grillot, and Bénédicte Votan, from ARCAGY–GINECO. We thank all the investigators who participated in the trial), all members of the study teams, pharmacists, pathologists, biologists, and study nurses from all the investigational sites. The medical writing support was provided by Sarah Mackenzie, PhD. The Sponsor (Arcagy-GINECO) was involved in each step of the trial design, and was entirely responsible for data collection. It covered the costs of the data analysis and manuscript writing. The authors did not receive any grant funding. The funding of the trial was covered by AstraZeneca through the Sponsor. No grant number is applicable.

## Author contributions

GF conceived and designed this study. GF, AF, OT, FS, CA, AL, CD-P, MF, LG, and ML enrolled patients and collected data. CL-J performed the statistical analyses, AC and JL analyzed biomarker data, and all authors participated in data interpretation. GF drafted the manuscript and all authors reviewed and revised it. The final version was approved by all authors.

## Competing interests

**GF**: honoraria (AstraZeneca, MSD, BMS, Lilly, GSK, Seagen, Daiichi-Sankyo); **AF**: honoraria (AstraZeneca, GSK, Clovis), meeting support (AstraZeneca, GSK, Pharma Mar), leadership in other board (GSK); **OT**: honoraria (Roche, Pfizer, Novartis-Sandoz, Lilly, MSD, AstraZeneca, Pierre Fabre, Seagen, Daiichi-Sankyo, Gilead, Eisai, Menarini-Stemline), meeting support (Roche, Pfizer, Novartis-Sandoz, Lilly, MSD, AstraZeneca, Seagen, Daiichi-Sankyo, Gilead), consulting/board (Roche, Pfizer, Novartis-Sandoz, Lilly, MSD, AstraZeneca, Pierre Fabre, Seagen, Daiichi-Sankyo, Gilead, Eisai); **FS**: consulting (AstraZeneca, MSD, GSK-Tesaro), honoraria (AstraZeneca, MSD, GSK-Tesaro), meeting support (AstraZeneca, MSD, GSK-Tesaro, Roche), **CA** grants (GSK), honoraria (GSK, Clovis Oncology, AstraZeneca), meeting support (GSK); **AL**: honoraria (AstraZeneca, GSK, Clovis), consulting (AstraZeneca, GSK, Clovis, MSD, Merck Serono, Ability, Zentalis),funded research (AstraZeneca, GSK, Clovis, MSD, Ability, Agenus, Iovance, Sanofi, Roche, OSEimmuno, BMS); **MF**: honoraria (AstraZeneca, GSK), consulting/board (AstraZeneca, GSK), **LG**: honoraria (AstraZeneca), meeting support (AstraZeneca). The remaining authors declare no competing interests (CL-J, CD, JL, AC, ML).

## Additional information

[1]Department of Medical Oncology, Lyon 1 University, Lyon, France. [2]GINECO (Groupe d'Investigateurs Nationaux pour l'Etude des Cancers de l'Ovaire, Paris, France. [3]Institut de Cancérologie des HCL, Lyon, France. [4]Department of Medical Oncology - Gynecological Tumors, Institut Bergonié, Bordeaux, France. [5]Medical Oncology, Centre Léon Bérard, Lyon, France. [6]EMR 3738, UFR Lyon-Sud, Université Lyon1, Lyon, France. [7]Biostatistics and Bioinformatics Department, Hospices Civils de Lyon, Lyon, France. [8]Department of Biochemistry and Molecular Biology, Hospices Civils de Lyon, Lyon, France. [9]Department of Medical Oncology, Groupe Hospitalier Diaconesses Croix Saint-Simon, Paris, France. [10]Gynecologic Oncology Department, Centre Oscar Lambret, Lille, France. [11]Oncology Department, Institut Gustave Roussy, Villejuif, France. [12]Medical Oncology, Institut Curie Saint Cloud, Paris, France. [13]Department of Medical Oncology, Institut du Cancer de Montpellier, Montpellier, France. [14]Medical Oncology, Institut Claudius Regaud IUCT-Oncopole, Toulouse, France. [15]Medical Oncology, Institut de Cancérologie du CHUSE, Saint-Etienne, France.
✉e-mail: gilles.freyer@univ-lyon1.fr

