## [Peer Review File · Nature Communications]

Bevacizumab, Olaparib, and Durvalumab in Patients with Relapsed Ovarian Cancer: A Phase II Clinical TrialREVIEWER COMMENTS

Reviewer #1 (Remarks to the Author): with expertise in ovarian cancer, therapy

The study by Freyer and colleagues assessed combination of bevacizumab, olaparib, and durvalumab in 74 patients with relapsed platinum-sensitive or platinum-resistant ovarian cancer. The study demonstrated a 3-month non-progression rate of 69.8% for platinum-resistant patients and 6-month non-progression rate of 43.8% for platinum-sensitive patients, meeting criteria for rejection of null hypothesis for the platinum-resistant, but not platinum-sensitive cohort.

Survival correlated with favorable KELIM-B constant and baseline immune inflammatory signature in tumors.

This is a well-presented study and the manuscript is well-written. The study demonstrated promising disease control rate, particularly in the platinum-resistant cohort despite significant exposure to prior bevacizumab.

I have several suggestions for the authors.

1. Inclusion of the numbers of patients treated in each cohort would be useful in the abstract.
2. Minor: discussion of KELIM modeling in methods is rather extensive and can be put into supplementary
3. Instead, the study could use a more detailed discussion of the tumor inflammation signature. At present, it is not even clear whether these were biopsies or archival tissues. If archival, it would be useful to know whether archival tissues were pre- or post- neoadjuvant chemo (as chemo could affect the inflammatory signature).
4. Related to above, absolutely no data are shown on the Nanostring signature, making it difficult for the reviewer to judge the conclusions reached by the authors. These data need to be included.

Reviewer #2 (Remarks to the Author): with expertise in ovarian cancer, therapy

Thank you for giving me the opportunity to review the manuscript. This topic (triplet

therapy: PARPi, Antiangiogenic agent, immune checkpoint inhibitor) is very interesting in the fields of ovarian cancer. Several randomized phase 3 studies are ongoing and are not published yet. To look into the data of completed phase 2 studies (MEDIOLA NCT02734004, OPEB-01 NCT04361370, and BOLD) gives the readers the insight about the applicability of triplet therapy in ovarian cancer.

Major comments

1. Interestingly, the authors suggest that triplet combination as a therapy seems more promising in platinum resistant rather than in platinum sensitive (mPFS 4.1m in platinum resistant / mPFS 4.9 m in platinum sensitive). It was different findings from triplet therapy in MEDIOLA study showing mPFS of 15m.

When we look at the reason, we can easily find that BOLD study includes 41% of prior PARPi user in the study. MEDIOLA study excludes prior PARPi users and prior IO users.

It would be necessary to show the efficacy of subgroup analysis according to prior PARPi users or not. Please specify how many patients had progressed with prior PARPi.

2. I would like to recommend addition of following discussion.

Please compare the efficacy of platinum sensitive / PARPi naïve patients from BOLD with MEDIOLA population.

Please compare the efficacy of platinum sensitive / PARPi resistant patients from BOLD with OREO population.

3. The most difficult thing about the interpretation of the research is heterogeneous population in small study population. Platinum resistant/ platinum sensitive, BRCAm/ BRCAwt, prior Antiangiogenic agent user or not/ prior PARPi user or not

It would be better to suggest the outcomes according to each group.

4. Figure 2, please specify prior PARPi use or not, prior AAI use or not from each patient.

Please specify the reasons for discontinuation from each patient. PD or withdraw or unacceptability from drug

Please add a vertical line at 3 months (for platinum resistant) and 6 months (for platinum sensitive). Better understanding from the figure.

5. Table 1, Please show the treatment free interval (platinum) / treatment free interval (biologic agent) in both cohorts. Median (range) of TFI is necessary to understand the cohort.

6. For exploratory analysis, I recommend that the authors suggest PD-L1 status and HRD

score.

7. Important conclusion is triplet is safe and well tolerated by patients with majority were acceptable. Safety section is valuable.

Is there any grade 4 events? If so, please specify.

It will be relevant for the readers to see the % of dose discontinuation individually and all by patients.

Please specify the irAEs separately.

Please specify the reason why 12 patients withdraw the treatment.

8. Discussion, please comment the efficacy of triplet therapy in patients who progressed in prior PARPi. It would be different activity of triplet maintenance according to PARPi naïve or resistant.

Minor comments

Line 38 “high-grade epithelial relapsed advanced ovarian cancer” seems unfamiliar. Please change it.

Line 144, any other exploratory analysis results such as PD-L1 or HRD?

Line 184, please compare the outcomes with other studies using doublet in platinum-resistant setting. Such as NRG-GY005 or KGOG 3045

Line 290, was durvalumab started from C1?

Line 375, It is necessary to check whether there are any statistical problems.

Line 381, do you mean that additional 18 patients did not receive treatment and was not included in the analysis?

Reviewer #3 (Remarks to the Author): with expertise in ovarian cancer, therapy

In this paper, Freyer et al. reported the findings of a single-arm, phase II trial of 3-drug combination with PARP inhibitor (PARPi) olaparib, anti-PDL1 durvalumab and bevacizumab in recurrent high-grade epithelial ovarian cancer. The study has two independent cohorts including platinum-resistant and platinum-sensitive recurrent diseases.

The primary endpoint is the non-progression rate at 3 months and at 6 months for platinum-resistant and platinum-sensitive disease, respectively.

Secondary endpoints are CA-125 decline (according to KELIM-B), PFS, OS, tumor response,

and safety evaluation. Translational objectives included the correlation of immune-related biomarkers with efficacy outcomes. Tumor response was evaluated by RECIST every 6 weeks until progression and serum CA-125 levels were determined every 6 weeks.

Overall, the manuscript is well-written and reports the findings of combination of 3-pathway (DNA damage repair, immune checkpoint and VEGF/VEGFR signaling) inhibition.

However, the idea of 3-pathway inhibition is not new and the findings are not strong enough to advance the scientific and clinical ovarian cancer research fields given 1) a single arm pilot study with one-stage design and lack of control group which makes difficult to interpret the clinical findings and 2) lack of proof-of-concept or novel biomarker findings due to using archival tissue samples instead of fresh biopsies and by evaluating only limited list of potential biomarkers.

Major comments:

1. Clinical findings

Albeit the statistically significant results in the primary outcome for the platinum-resistant cohort, the clinical meaningfulness of mPFS results disputably. The AURELIA study demonstrated a PFS of 6.7 months with the addition of bevacizumab to the standard chemotherapy, which is much higher than the mPFS of 4.1 months reported in the present study (Pujade-Lauraine E, et al. J Clin Oncol. 2014).

Furthermore, recent trials investigating the efficacy of novel and bevacizumab-free strategies showed higher mPFS than this triplet combination though we can't compare this study with others head-to-head (mPFS of 5.6 months with intermittent relacorilant plus nab-paclitaxel, CORT125134- 552 study, Colombo N. et al. ESMO Congress. Virtual; 2021.

Abstract 7210; mPFS of 4.6 months with adavosertib plus gemcitabine (Lheureux S, et al. Lancet. 2021:281-292; mPFS of 4.3 months and 5.5 months assessed by investigator and in the BICR efficacy evaluable population respectively with the use of Mirvetuximab Soravtansine in the SORAYA study, Matulonis UA et al., J Clin Oncol. 2023).

On the other end, it is hard to interpret the relevance of the clinical findings from the platinum-sensitive cohort, as the trial did not meet the planned number of patients to be

enrolled. However, the exploratory mPFS of 4.9 months appears disappointing.

2. Correlative study evaluation

Although the identification of biomarkers of efficacy with this triplet regimen might be challenging due to the concurrent inhibition of 3 different pathways, the soundness of the chosen predictive biomarkers is scarce, as they do not add any novelty to ovarian cancer and immune-oncology fields.

Both the KELIM score in ovarian cancer and the 18-gene tumor inflammation signature (TIS) in the immune-oncology field are widely studied biomarkers to predict therapeutic responses, thus being outdated. KELIM-PARP has already been studied as a marker to identify patient prognosis, complementary to platinum-sensitivity and HRD status. It would have been more appealing to test a new biomarker capable of predicting response to the triple combination rather than related to the DNA repair pathway only. On the other hand, TIS is a well know biomarker of immune checkpoint inhibitor response, which measures the pre-existing adaptive immune resistance phenotype within the tumor microenvironment but lacks information about the innate immune system involvement in treatment resistance/response.

Also, the choice of using archival formalin-fixed paraffin-embedded (FFPE) tissue samples challenge the understanding of biology and limits the interpretation of omics data due to current technical issues with archival tissue. Furthermore, stored materials instead of fresh samples do not allow a comprehensive insight into the current tumor biology and the collection of paired samples to underscore dynamic changes within tumor tissues during cancer treatments. Indeed, the Surgery Committee of the Society for Immunotherapy of Cancer (SITC) recommended the preferred use of fresh tissues for both genomic and immune studies [Gastman et al., Defining best practices for tissue procurement in immunology clinical trials: a consensus statement from the Society for Immunotherapy of Cancer Surgery Committee. *J Immunother Cancer*. 2020].

Those observations suggest that comprehensive mechanistic and/or correlative studies from samples are lacking, therefore not meeting the quality required by Nature Communication.

3. Study population

Patient populations are slightly different between the two cohorts. In particular, it is worth noting that just one-third of the platinum-resistant population (32%) had previously received PARPi. In contrast, over half of the platinum-sensitive population was previously exposed to PARPi treatment (52%). Despite just being thought-provoking, it can be speculated that the difference in the extent of prior treatment might have affected the responses of this 3-drug combination between the two cohorts. Besides, with the increasing use of PARPi in the upfront maintenance setting, the majority of secondary platinum-resistant patients will likely be exposed to PARPi, and the percentage of the previously treated patients in this study is too small to draw any conclusion about the PARPi re-exposure given with this combination therapy.

4. Correlative studies: prognostic versus predictive biomarkers

There needs to be clarity on the interpretation of predictive and prognostic biomarkers across the manuscript. The authors defined both KELIM and TIS as prognostic biomarkers. Also, in the discussion, it is stated that “The identification of prognostic biomarkers is an important aspect of current drug development programs. KELIM™ estimation is a recently developed tool which may be exploited in ovarian cancer setting to identify patients who are more likely to benefit from treatment, notably with respect to PFS and OS with bevacizumab maintenance treatment in the first-line setting”. However, by definition, a prognostic biomarker provides information about patient overall cancer outcomes, regardless of therapy, while a predictive biomarker gives information about the effect of a therapeutic intervention. Even if the gold standard for the identification of predictive biomarkers is a randomized trial, a biomarker used to identify patients “who are more likely to benefit from treatment” has a predictive value rather than prognostic.

5. Statistical considerations

The authors performed exploratory efficacy analyses according to BRCA1/2 status. Yet, they only described the absolute differences without a statistical comparison. For example, the difference in ORR is highlighted between the two groups, especially for the platinum-resistant cohort. However, it would be informative to statistically compare the two groups to support that claim by providing odds ratio and p-value despite the small sample

size. Interestingly, PFS according to BRCA1/2 status appeared similar between the two groups.

Translational endpoints are all exploratory and unpowered, thus data are only hypothesis generating and should be interpreted with caution which the authors should add to the study limitations.

Minor concerns:

1. Introduction

The authors are encouraged to rephrase the following statement: “The backbone of standard therapy after relapse for patients with the platinum-sensitive disease is a platinum-based combination, administered until the development of platinum resistance”. It seems inaccurate and may confuse the reader, as platinum-based chemotherapy is not given until disease progression. Please clarify.

2. Figure 2. dagger (+) representing PD is shown in the 3rd, and 17th patients from the top (platinum resistant relapse patients) who seem to have stayed on the study longer than 5 months or with ongoing response, please double check.

3. Safety considerations

- The authors described only the “most common AEs” (text) or “Main AEs” (table 4), but they did not specify the threshold to define them (frequency >10%? >20%?).
- No treatment-related adverse events (TRAEs), treatment interruptions, or dose reductions were described. The authors are strongly encouraged to clarify their occurrence, if any.

Reviewer #4 (Remarks to the Author): with expertise in biostatistics, clinical trial study design

I have reviewed the article “Bevacizumab, Olaparib, and Durvalumab in Patients with Relapsed Ovarian Cancer: the GINECO BOLD Study”. This study investigated the use of triple combination in advanced ovarian patients with platinum-resistance relapse and platinum-

sensitive relapse. The authors reported that non-progression rate at 3 months of 69.8% was achieved in the platinum-resistance cohort. I have the following comments.

1. Primary endpoint:

Please provide the data cutoff date. As the primary endpoint is non-progression rate estimated by the Kaplan-Meier method, please provide Kaplan-Meier curves with lines showing the non-progression rates at 3 months in the platinum-resistance cohort and 6 months in the platinum-sensitive cohort. The method section lists some censoring criteria. Is there any patient censored for the primary endpoint? The censored patients should be shown in all Kaplan-Meier curves.

2. Objective response:

In Table 2, the objective response ($11/39=28\%$) is reported in 39 evaluable patients per RECIST in the platinum-resistance cohort. However, the objective response ($12/33=36\%$) is reported in 33 ITT patients in the platinum-sensitive cohort. Please use the same population in both cohorts and harmonize the text with the table. What are the reasons of the 3 patients being non-evaluable for RECIST and 2 patients being non-evaluable for irRECIST?

3. BRCA 1/2 mutation:

Four patients had BRCA 1/2 mutations in the platinum-resistance cohort. The authors compared the non-progression rates in 4 mutant vs. 37 wild-type. However, the number of wild-type should be the number of available mutation status minus the number of mutant. Please clarify the number of wild-type and the population used in the comparison. Same situation happens in the platinum-sensitive cohort.

4. Swimmer plot:

Figure 2 only shows 71 patients. Are these 3 patients the same as the 3 patients who are not evaluable for objective response? What are the reasons that they are not in the swimmer plot?

In the legend, does the question mark under the BRCA status mean no mutation or not available? There should be 4 categories: not available, no mutation, wild-type and mutant. The ongoing should not be placed in BRCA status.

The method section states that treatment was administered until progression, unacceptable toxicity or withdrawal of consent. In the platinum-resistance cohort, what were the reasons of the two patients with PD received ongoing treatments?

The numbers of mutant and wild-type do not match with Table 1.

Please provide the vital status at the end of each bar and change the months of treatment in the x-axis from the increment of 5 months to 3 months.

5. Protocol:

When it was found that 28 platinum-resistance patients and 28 platinum-sensitive patients were accrued in the data routine monitoring, 13 more platinum-resistance patients and 5 more platinum-sensitive were recruited given the increasing success of the protocol and the Steering Committee's recommendation. Was the protocol amended for this change? What was the reason of not completing the recruitment in the platinum-sensitive cohort?

REVIEWER COMMENTS

Reviewer #1 (Remarks to the Author): with expertise in ovarian cancer, therapy

The study by Freyer and colleagues assessed combination of bevacizumab, olaparib, and durvalumab in 74 patients with relapsed platinum-sensitive or platinum-resistant ovarian cancer. The study demonstrated a 3-month non-progression rate of 69.8% for platinum-resistant patients and 6-month non-progression rate of 43.8% for platinum-sensitive patients, meeting criteria for rejection of null hypothesis for the platinum-resistant, but not platinum-sensitive cohort.

Survival correlated with favorable KELIM-B constant and baseline immune inflammatory signature in tumors.

This is a well-presented study and the manuscript is well-written. The study demonstrated promising disease control rate, particularly in the platinum-resistant cohort despite significant exposure to prior bevacizumab.

I have several suggestions for the authors.

1. Inclusion of the numbers of patients treated in each cohort would be useful in the abstract. **Author response:** added
2. Minor: discussion of KELIM modeling in methods is rather extensive and can be put into supplementary. **Author response:** Moved to a separate Supplementary Methods/Data file
3. Instead, the study could use a more detailed discussion of the tumor inflammation signature. At present, it is not even clear whether these were biopsies or archival tissues. If archival, it would be useful to know whether archival tissues were pre- or post-neoadjuvant chemo (as chemo could affect the inflammatory signature). **Author response:** Per protocol, a baseline tumor biopsy sample - FFPE archival or fresh – obtained within 3 months prior to treatment start and after the last chemotherapy administration was mandatory. This has been further clarified in the methods.
4. Related to above, absolutely no data are shown on the Nanostring signature, making it difficult for the reviewer to judge the conclusions reached by the authors. These data need to be included. **Author response:** Additional baseline and efficacy data have been added to the results /supplementary data.

Reviewer #2 (Remarks to the Author): with expertise in ovarian cancer, therapy

Thank you for giving me the opportunity to review the manuscript. This topic (triplet therapy: PARPi, Antiangiogenic agent, immune checkpoint inhibitor) is very interesting in the fields of ovarian cancer. Several randomized phase 3 studies are ongoing and are not published yet. To look into the data of completed phase 2 studies (MEDIOLA NCT02734004, OPEB-01 NCT04361370, and BOLD) gives the readers the insight about

the applicability of triplet therapy in ovarian cancer.

Major comments

1. Interestingly, the authors suggest that triplet combination as a therapy seems more promising in platinum resistant rather than in platinum sensitive (mPFS 4.1m in platinum resistant / mPFS 4.9 m in platinum sensitive). It was different findings from triplet therapy in MEDIOLA study showing mPFS of 15m.

When we look at the reason, we can easily find that BOLD study includes 41% of prior PARPi user in the study. MEDIOLA study excludes prior PARPi users and prior IO users. It would be necessary to show the efficacy of subgroup analysis according to prior PARPi users or not. Please specify how many patients had progressed with prior PARPi. **Author response:** the number of patients exposed to prior PARPi is indicated by cohort and drug in the results text and Table 1 (32% for the platinum-resistant cohort and 52% for the platinum-sensitive cohort). We have added efficacy data in supplementary results and mention the population in the discussion.

2. I would like to recommend addition of following discussion.

Please compare the efficacy of platinum sensitive / PARPi naïve patients from BOLD with MEDIOLA population.

Please compare the efficacy of platinum sensitive / PARPi resistant patients from BOLD with OREO population. **Author response:** We note that for both of these studies, neither have been published with final data in peer-reviewed journals. We have added text in the discussion to provide context for the initial data reported at the most recent conferences.

3. The most difficult thing about the interpretation of the research is heterogeneous population in small study population. Platinum resistant/ platinum sensitive, BRCAm/ BRCAwt, prior Antiangiogenic agent user or not/ prior PARPi user or not

It would be better to suggest the outcomes according to each group. **Author response:** We agree that this heterogeneity limits the interpretation of the results, however on the other hand BOLD is a pragmatic study that included patients as they presented in the real-world oncology practice. Our purpose was to evaluate the efficacy of the triplet combination in – more or less – heavily pretreated patients, to obtain a signal showing a more promising efficacy than any of the 3 drugs alone. In addition, the sample sizes are too small to support outcomes by these subgroup analyses – as noted in the discussion where BRCA is mentioned and in the study limitations section. We therefore respectfully disagree with this recommendation.

4. Figure 2, please specify prior PARPi use or not, prior AAI use or not from each patient.

Author response: See new Figure 3 swimmer plot (previously Figure 2) with the addition of a symbol 'P'.

Please specify the reasons for discontinuation from each patient. PD or withdraw or unacceptability from drug **Author response:** The authors consider adding the reason for treatment discontinuation to the swimmer for each patient would detract from

readability of this efficacy driven figure. We have added in the text the number of patients discontinuing due to AEs.

Please add a vertical line at 3 months (for platinum resistant) and 6 months (for platinum sensitive). Better understanding from the figure. **Author response:** The new figure has intervals of 3 months making this easy for the reader to follow.

5. Table 1, Please show the treatment free interval (platinum) / treatment free interval (biologic agent) in both cohorts. Median (range) of TFI is necessary to understand the cohort. **Author response:** The platinum-free interval for each cohort has been added to Table 1. Data were not systematically collected for biological agents such as bevacizumab.

6. For exploratory analysis, I recommend that the authors suggest PD-L1 status and HRD score. **Author response:** We agree this would be interesting to explore, however the data from the translational research were limited restricting analyses and rendering outcomes speculative. A summary of baseline HRD data has been added to supplementary data. Baseline PD-L1 data were not scored as percent positive cells and are thus not presented.

7. Important conclusion is triplet is safe and well tolerated by patients with majority were acceptable. Safety section is valuable.

Is there any grade 4 events? If so, please specify.

It will be relevant for the readers to see the % of dose discontinuation individually and all by patients.

Please specify the irAEs separately.

Please specify the reason why 12 patients withdraw the treatment.

Author response: Grade 4 events have been added to the safety results.

The safety section already reports that 12 patients discontinued due to AEs; we have added details of AE type and which drugs were discontinued.

Immune-related AEs are called out in detail in the text results and discussion and we prefer not to separate them out in the table.

8. Discussion, please comment the efficacy of triplet therapy in patients who progressed in prior PARPi. It would be different activity of triplet maintenance according to PARPi naïve or resistant. **Author response:** Text has been added to the discussion.

Minor comments

Line 38 "high-grade epithelial relapsed advanced ovarian cancer" seems unfamiliar.

Please change it. **Author response:** changed to relapsed high-grade advanced ovarian cancer

Line 144, any other exploratory analysis results such as PD-L1 or HRD? **Author response:** We agree this is an interesting aspect, however the data from the translational research

were limited. A summary of HRD data has been added to supplementary data. PD-L1 data were not scored as percent positive cells and are thus not presented.

Line 184, please compare the outcomes with other studies using doublet in platinum-resistant setting. Such as NRG-GY005 or KGOG 3045. **Author response:** To our knowledge, the results of the NRG GY005 trial have not yet been published. In the KCOG 3045 trial, 14 patients with HRD-positive, platinum-resistant ovarian cancer received olaparib plus durvalumab. The authors report an ORR of 43% but with a large confidence interval. It is difficult to compare their results with those of the BOLD study, where our patients were not selected on the basis of HRD status.

Line 290, was durvalumab started from C1? **Author response:** yes – clarified in text

Line 375, It is necessary to check whether there are any statistical problems. **Author response:** No, we do not consider the null hypothesis would have been rejected by including 10 additional patients in the platinum-sensitive cohort. The addition of extra patients in the platinum-resistant cohort likely strengthened the power of the statistical analysis.

Line 381, do you mean that additional 18 patients did not receive treatment and was not included in the analysis? **Author response:** No the 18 patients were already included and treated – there was no further treatment available to manage additional patients beyond these 18; text has been clarified.

Reviewer #3 (Remarks to the Author): with expertise in ovarian cancer, therapy

In this paper, Freyer et al. reported the findings of a single-arm, phase II trial of 3-drug combination with PARP inhibitor (PARPi) olaparib, anti-PDL1 durvalumab and bevacizumab in recurrent high-grade epithelial ovarian cancer. The study has two independent cohorts including platinum-resistant and platinum-sensitive recurrent diseases.

The primary endpoint is the non-progression rate at 3 months and at 6 months for platinum-resistant and platinum-sensitive disease, respectively.

Secondary endpoints are CA-125 decline (according to KELIM-B), PFS, OS, tumor response, and safety evaluation. Translational objectives included the correlation of immune-related biomarkers with efficacy outcomes. Tumor response was evaluated by RECIST every 6 weeks until progression and serum CA-125 levels were determined every 6 weeks.

Overall, the manuscript is well-written and reports the findings of combination of 3-pathway (DNA damage repair, immune checkpoint and VEGF/VEGFR signaling) inhibition. However, the idea of 3-pathway inhibition is not new and the findings are not strong enough to advance the scientific and clinical ovarian cancer research fields given 1) a single arm pilot study with one-stage design and lack of control group which makes

difficult to interpret the clinical findings and 2) lack of proof-of-concept or novel biomarker findings due to using archival tissue samples instead of fresh biopsies and by evaluating only limited list of potential biomarkers.

Major comments:

1. Clinical findings

Albeit the statistically significant results in the primary outcome for the platinum-resistant cohort, the clinical meaningfulness of mPFS results disputably. The AURELIA study demonstrated a PFS of 6.7 months with the addition of bevacizumab to the standard chemotherapy, which is much higher than the mPFS of 4.1 months reported in the present study (Pujade-Lauraine E, et al. J Clin Oncol. 2014). **Author response:** Whilst we agree on the face of it with this comment, we note that the patient population in the AURELIA study had a better prognosis with 60% in first relapse, 40% in second, and only 8% had previously received bevacizumab. Whereas, our BOLD population had a median of 3 prior lines of therapy and 85% had received prior bevacizumab. This is stated in the first paragraph of the discussion.

Furthermore, recent trials investigating the efficacy of novel and bevacizumab-free strategies showed higher mPFS than this triplet combination though we can't compare this study with others head-to-head (mPFS of 5.6 months with intermittent relacorilant plus nab-paclitaxel, CORT125134- 552 study, Colombo N. et al. ESMO Congress. Virtual; 2021. Abstract 7210; mPFS of 4.6 months with adavosertib plus gemcitabine (Lheureux S, et al. Lancet. 2021:281-292; mPFS of 4.3 months and 5.5 months assessed by investigator and in the BICR efficacy evaluable population respectively with the use of Mirvetuximab Soravtansine in the SORAYA study, Matulonis UA et al., J Clin Oncol. 2023).

On the other end, it is hard to interpret the relevance of the clinical findings from the platinum-sensitive cohort, as the trial did not meet the planned number of patients to be enrolled. However, the exploratory mPFS of 4.9 months appears disappointing. **Author response:** Whilst we agree that the PFS may be considered as disappointing, however 1) in the above mentioned studies, the populations were less pretreated than the BOLD population, and thus of better prognosis (the poor prognosis of our more pretreated population is noted in the discussion) and 2) the median OS in BOLD is close to 19 months which is encouraging and likely reflects the efficacy of immunotherapy.

2. Correlative study evaluation

Although the identification of biomarkers of efficacy with this triplet regimen might be challenging due to the concurrent inhibition of 3 different pathways, the soundness of the chosen predictive biomarkers is scarce, as they do not add any novelty to ovarian cancer and immune-oncology fields. **Author response:** We performed a broad biomarker analysis; data are presented data for TIS, as this was the only biomarker analyzed that proved significant. We have added data in supplementary data for the

other biomarkers analyzed. Note that biomarker data were not available for all patients and these results are exploratory – we have noted this in the discussion.

Both the KELIM score in ovarian cancer and the 18-gene tumor inflammation signature (TIS) in the immune-oncology field are widely studied biomarkers to predict therapeutic responses, thus being outdated. KELIM-PARP has already been studied as a marker to identify patient prognosis, complementary to platinum-sensitivity and HRD status.

Author response: KELIM was studied in the VELIA trial where the PARP inhibitor was given in combination with chemotherapy. To date, BOLD is the first study to show the predictive value of KELIM in a chemotherapy-free regimen. Moreover, data regarding TIS in ovarian cancer are very limited. We have added a statement to this effect to the discussion.

It would have been more appealing to test a new biomarker capable of predicting response to the triple combination rather than related to the DNA repair pathway only. On the other hand, TIS is a well know biomarker of immune checkpoint inhibitor response, which measures the pre-existing adaptive immune resistance phenotype within the tumor microenvironment but lacks information about the innate immune system involvement in treatment resistance/response. **Author response:** We agree with this comment but note that our analysis was performed on tissue obtained prior to treatment.

Also, the choice of using archival formalin-fixed paraffin-embedded (FFPE) tissue samples challenge the understanding of biology and limits the interpretation of omics data due to current technical issues with archival tissue. Furthermore, stored materials instead of fresh samples do not allow a comprehensive insight into the current tumor biology and the collection of paired samples to underscore dynamic changes within tumor tissues during cancer treatments. Indeed, the Surgery Committee of the Society for Immunotherapy of Cancer (SITC) recommended the preferred use of fresh tissues for both genomic and immune studies [Gastman et al., Defining best practices for tissue procurement in immuno-oncology clinical trials: a consensus statement from the Society for Immunotherapy of Cancer Surgery Committee. J Immunother Cancer. 2020].

Those observations suggest that comprehensive mechanistic and/or correlative studies from samples are lacking, therefore not meeting the quality required by Nature Communication. **Author response:** Whilst we agree that fresh frozen biopsies are optimal, FFPE biopsies collected within 3 months prior to treatment start and after the last chemotherapy administration, as noted in the methods, is widely used routine practice for biomarker analyses, notably in the context of exploratory analyses, given that fresh frozen samples represent logistical challenges and are costly (requiring dry ice and immediate dispatch, etc.) making them inaccessible to a large proportion of cancer centers.

3. Study population

Patient populations are slightly different between the two cohorts. In particular, it is worth noting that just one-third of the platinum-resistant population (32%) had

previously received PARPi. In contrast, over half of the platinum-sensitive population was previously exposed to PARPi treatment (52%). Despite just being thought-provoking, it can be speculated that the difference in the extent of prior treatment might have affected the responses of this 3-drug combination between the two cohorts. **Author response:** We agree but it is very unlikely that PARPi monotherapy would have proven substantial efficacy in PRR patients with no *BRCA* mutation.

Besides, with the increasing use of PARPi in the upfront maintenance setting, the majority of secondary platinum-resistant patients will likely be exposed to PARPi, and the percentage of the previously treated patients in this study is too small to draw any conclusion about the PARPi re-exposure given with this combination therapy. **Author response:** We agree that conclusions cannot be drawn from our single arm populations. We have elaborated this in the discussion limitations. We also consider that based on our results, the BOLD triplet merits evaluation in this setting, as stated in our concluding paragraph.

4. Correlative studies: prognostic versus predictive biomarkers

There needs to be clarity on the interpretation of predictive and prognostic biomarkers across the manuscript. The authors defined both KELIM and TIS as prognostic biomarkers. Also, in the discussion, it is stated that “The identification of prognostic biomarkers is an important aspect of current drug development programs. KELIMTM estimation is a recently developed tool which may be exploited in ovarian cancer setting to identify patients who are more likely to benefit from treatment, notably with respect to PFS and OS with bevacizumab maintenance treatment in the first-line setting”. However, by definition, a prognostic biomarker provides information about patient overall cancer outcomes, regardless of therapy, while a predictive biomarker gives information about the effect of a therapeutic intervention. Even if the gold standard for the identification of predictive biomarkers is a randomized trial, a biomarker used to identify patients “who are more likely to benefit from treatment” has a predictive value rather than prognostic.

Author response: We thank the reviewer for pointing this out. By definition, a model-based parameter reflecting the CA 125 decrease under treatment (KELIM) and its influence on OS is both prognostic and predictive. KELIM was shown to be strongly correlated to survival in first line (prognostic) and to the efficacy of chemotherapy plus bevacizumab (predictive in ICON 7 and GOG 218 patients) and of chemotherapy plus veliparib (predictive – in VELIA patients). To date, most studies in oncology report “prognostic” parameters coming from patients who are under treatment, since these parameters influence a patient’s OS, and “predictive” parameters when these parameters are strongly linked to chemotherapy efficacy (e.g., ORR). We agree that the definition of a “prognostic” parameter in this setting is not absolutely pure.

5. Statistical considerations

The authors performed exploratory efficacy analyses according to BRCA1/2 status. Yet, they only described the absolute differences without a statistical comparison.

For example, the difference in ORR is highlighted between the two groups, especially for the platinum-resistant cohort. However, it would be informative to statistically compare the two groups to support that claim by providing odds ratio and p-value despite the small sample size. Interestingly, PFS according to BRCA1/2 status appeared similar between the two groups. **Author response:** We have modified the text to remove comparison, and instead highlight the small numbers and that these data are exploratory. Given the small number of *BRCA* mutated patients and the lack of randomisation, it is not feasible to perform statistical comparisons that are clinically meaningful and methodologically acceptable.

Translational endpoints are all exploratory and unpowered, thus data are only hypothesis generating and should be interpreted with caution which the authors should add to the study limitations. **Author response:** We note in the results section that the translational data are exploratory and this point is also noted in the limitations section – we have further clarified.

Minor concerns:

1. Introduction

The authors are encouraged to rephrase the following statement: “The backbone of standard therapy after relapse for patients with the platinum-sensitive disease is a platinum-based combination, **for repeated lines unless platinum resistance occurs.** ~~administered until the development of platinum resistance~~”. It seems inaccurate and may confuse the reader, as platinum-based chemotherapy is not given until disease progression. Please clarify. **Author response:** The sentence has been modified as suggested.

2. Figure 2. dagger (+) representing PD is shown in the 3rd, and 17th patients from the top (platinum resistant relapse patients) who seem to have stayed on the study longer than 5 months or with ongoing response, please double check. **Author response:** The study protocol allows for investigators to continue treatment in the event of progression, if he/she considered that the treatment was beneficial for the patient. In the case of these 2 patients, no further progression was observed in subsequent evaluations and the clinical status remained favorable.

3. Safety considerations

- The authors described only the “most common AEs” (text) or “Main AEs” (table 4), but they did not specify the threshold to define them (frequency >10%? >20%?). **Author response:** Thank you for pointing this out – the threshold has been specified and table corrected.

- No treatment-related adverse events (TRAEs), treatment interruptions, or dose reductions were described. The authors are strongly encouraged to clarify their occurrence, if any. **Author response:** for a single-arm cohort study we consider that

reporting all AEs irrespective of relationship is more appropriate than reporting related AEs.

Reviewer #4 (Remarks to the Author): with expertise in biostatistics, clinical trial study design

I have reviewed the article "Bevacizumab, Olaparib, and Durvalumab in Patients with Relapsed Ovarian Cancer: the GINECO BOLD Study". This study investigated the use of triple combination in advanced ovarian patients with platinum-resistance relapse and platinum-sensitive relapse. The authors reported that non-progression rate at 3 months of 69.8% was achieved in the platinum-resistance cohort. I have the following comments.

1. Primary endpoint:

Please provide the data cutoff date. As the primary endpoint is non-progression rate estimated by the Kaplan-Meier method, please provide Kaplan-Meier curves with lines showing the non-progression rates at 3 months in the platinum-resistance cohort and 6 months in the platinum-sensitive cohort. The method section lists some censoring criteria. Is there any patient censored for the primary endpoint? The censored patients should be shown in all Kaplan-Meier curves. **Author response:** The cutoff date has been provided in the methods section. KM PFS provided for the resistant and sensitive cohorts (see new Figure 4). Censoring marks have been added to all KM files. Note that the p-value was corrected in the KM new Figure 4A (to match text).

2. Objective response:

In Table 2, the objective response (11/39=28%) is reported in 39 evaluable patients per RECIST in the platinum-resistance cohort. However, the objective response (12/33=36%) is reported in 33 ITT patients in the platinum-sensitive cohort. Please use the same population in both cohorts and harmonize the text with the table. **Author response:** Thank you for highlighting the incoherence. The data have been corrected in the table and text – ORR is reported in the evaluable population for all cases.

What are the reasons of the 3 patients being non-evaluable for RECIST and 2 patients being non-evaluable for irRECIST? **Author response:** Lack of measurable lesions. Patients could be included even if they had non-measurable lesions according to RECIST and irRECIST (detailed in patient eligibility in the methods).

3. BRCA 1/2 mutation:

Four patients had BRCA 1/2 mutations in the platinum-resistance cohort. The authors compared the non-progression rates in 4 mutant vs. 37 wild-type. However, the number of wild-type should be the number of available mutation status minus the number of mutant. Please clarify the number of wild-type and the population used in the comparison. Same situation happens in the platinum-sensitive cohort. **Author response:**

Data are presented for mutant versus wild-type plus missing; this has been clarified in the table.

4. Swimmer plot:

Figure 2 only shows 71 patients. Are these 3 patients the same as the 3 patients who are not evaluable for objective response? What are the reasons that they are not in the swimmer plot? **Author response:** The swimmer excludes the 3 patients who were not evaluable for response; Note Figure 2 is now Figure 3.

In the legend, does the question mark under the BRCA status mean no mutation or not available? There should be 4 categories: not available, no mutation, wild-type and mutant. The ongoing should not be placed in BRCA status. **Author response:** The symbols in the figure have been updated.

The method section states that treatment was administered until progression, unacceptable toxicity or withdrawal of consent. In the platinum-resistance cohort, what were the reasons of the two patients with PD received ongoing treatments? **Author response:** The study protocol allows for investigators to continue treatment in the event of progression, if he/she considered that the treatment was beneficial for the patient. In the case of these 2 patients, no further progression was observed in subsequent evaluations and the clinical status remained favorable.

The numbers of mutant and wild-type do not match with Table 1. **Author response:** the legend has been modified.

Please provide the vital status at the end of each bar and change the months of treatment in the x-axis from the increment of 5 months to 3 months. **Author response:** We prefer not to show vital status on the swimmer plot as this figure is busy and we would like to focus on efficacy data displayed.

5. Protocol:

When it was found that 28 platinum-resistance patients and 28 platinum-sensitive patients were accrued in the data routine monitoring, 13 more platinum-resistance patients and 5 more platinum-sensitive were recruited given the increasing success of the protocol and the Steering Committee's recommendation. Was the protocol amended for this change? **Author response:** The protocol was not amended, since additional patients had already been screened and approved for inclusion and were anticipating treatment. Given the severity of their disease, we considered it was not ethical to stop their inclusion for administrative reasons.

What was the reason of not completing the recruitment in the platinum-sensitive cohort? **Author response:** After discussion with the study methodologists and data simulations, we estimated that there was little chance of a major change to our results in the PSR cohort by including fewer than 15 – 20 patients. We therefore chose to include only the PRR patients who had already been screened by the investigators given that the screened patients were already approved for entry and that no further drug was available, as noted in the methods.

REVIEWER COMMENTS

Reviewer #1 (Remarks to the Author):

I appreciate the author responses to my comments. Inclusion of additional information around the TIS score provided some clarification, but also generated an additional concern:

The authors report in the manuscript that favorable TIS was associated with improved outcomes, which was more pronounced in each group. However, statistical support for this is not presented: Figures 4c-d just report the overall Log-Rank value; in supplementary figures 2 the threshold for TIS crosses 0, and in supplementary figure 3 no comparison is performed between the PR/CR and SD/PD groups. Any reference to “predictive value” of TIS in the text should thus be supported with a corresponding p-value/statistical comparison in the text and, if not significant, the language should be softened.

Reviewer #2 (Remarks to the Author):

Thank you for your response.

However, it is very hard to follow how the manuscript is changed per the reviewer's comments.

Reviewer #3 (Remarks to the Author):

The reviewer appreciates the authors' efforts to improve the quality of the manuscript. However, the quality of clinical and correlative study data may be statistically significant but not clinically meaningful in platinum resistant patients and will unlikely make an impact to this population.

Specifically, as the authors are well aware, clinical activity data - 4.1 months of median PFS of 3-drug combination in the platinum resistant population- are insufficient to move forward as a randomized phase 2/3 study for further clinical development. Also, provided correlative study findings appear to be suboptimal to provide new insights into this drug combination.

Lastly, the manuscript would benefit from getting further edits and English revision to be

suitable for publication.

Major comments:

1) To clarify the clinical significance (not only statistically significant) of this study, authors should also explain the contexts in this platinum resistant disease field, specifically concerning the efficacy outcomes of the major clinical trials in the similar setting. This should include at least AURELIA, SORAYA, MIRASOL, CORT125134- 552 studies as stated in the Reviewers' comments rebuttal (Major comments 1-2 of reviewer#3). It is essential to highlight the differences to understand the clinical impact of this trial compared to other reported studies given that most of them became the standard of care therapy options.

2) Figure 3:

(1) Please add a legend to specify in a different position that the symbol "P" refers to prior PARPi exposure. It can be misleading to include prior PARPi and PARPi naive without clarification in the best response legend.

(2) Please clarify why patients o continued the treatment beyond progression as explained in the rebuttal to reviewers#3 and #4.

(3) Please disclose the reason why the swimmer plot excluded three patients who were not evaluable for RECIST response.

Minor comments

1. Safety section:

(1) Please specify the threshold of the "most common AEs" in the main text according to table 4 (20%).

(2) Please edit the following sentence as suggested: withdrew from treatment due to an AE: including anemia (6 patients – olaparib)

2. Discussion section:

The following sentence (page 9) "This may be due to differences in the extent of prior treatment (patients had not received PARP inhibitors and immunotherapy approximately half of our cohort had received prior PARP inhibitors, whereas in MEDIOLA prior PARP inhibitor exposure was not permitted). " appears to be misinterpretation of the data

because the author's justification is for prior PARPi exposed group while the authors' data of 47% ORR seen in the 15 patients are for PARPi naïve group and 77% ORR in MEDIOLA study which is also for PARPi naïve group.

(4) Please add in the study limitations the use of archival tissues followed by an explanation about the reliability of the translational data.

Reviewer #4 (Remarks to the Author):

I have read the responses and revisions. The authors addressed the comments adequately and the manuscript is improved. I had the following minor comments.

1. On page 5, please insert “/ missing status” in the text since this phase was added to Tables 2 and 3.
2. In Supplementary Table 3, it would be better if row percentage is provided rather than the column percentage. So, the response could be compared between prior PARP inhibitor and no prior PARP inhibitor.
3. Please provide the method that was used to calculate the 95% CI of the objective response rate in Tables 2 and 3.
4. In the new Figure 3, ‘P’ indicates prior PARP inhibitor therapy which is not a response category. Please remove ‘P’ from Best response according RECIST.
5. Please change Figure 3D, 3F in the text to Figure 4D, 4F.
6. Thank you for providing the non-progression Kaplan-Meier curves of the two cohorts in the new Figure 2. It would help to understand the non-progression rates during the study. It appears that both cohorts had similar non-progression curves which might be due to the higher prior PARP inhibitor rate in the platinum sensitive cohort at baseline (52% vs 32%). In the platinum resistant cohort, the alternative hypothesis was that the non-progressive disease rate at 3 months was greater than 50% (historical control). For the platinum sensitive cohort, what was the rationale of choosing 65% non-progressive disease rate at 6 months as the null hypothesis? It was not stated neither in the protocol nor manuscript.

Reviewer #1 (Remarks to the Author):

I appreciate the author responses to my comments. Inclusion of additional information around the TIS score provided some clarification, but also generated an additional concern:

The authors report in the manuscript that favorable TIS was associated with improved outcomes, which was more pronounced in each group. However, statistical support for this is not presented: Figures 4c-d just report the overall Log-Rank value; in supplementary figures 2 the threshold for TIS crosses 0, and in supplementary figure 3 no comparison is performed between the PR/CR and SD/PD groups. Any reference to “predictive value” of TIS in the text should thus be supported with a corresponding p-value/statistical comparison in the text and, if not significant, the language should be softened.

Author response: we take the reviewers point and have softened the language in the results and removed the statement from the abstract.

Reviewer #3 (Remarks to the Author):

The reviewer appreciates the authors’ efforts to improve the quality of the manuscript. However, the quality of clinical and correlative study data may be statistically significant but not clinically meaningful in platinum resistant patients and will unlikely make an impact to this population. Specifically, as the authors are well aware, clinical activity data - 4.1 months of median PFS of 3-drug combination in the platinum resistant population- are insufficient to move forward as a randomized phase 2/3 study for further clinical development. Also, provided correlative study findings appear to be suboptimal to provide new insights into this drug combination. Lastly, the manuscript would benefit from getting further edits and English revision to be suitable for publication.

Author response: The manuscript has been extensively reviewed and edited by an English-native medical writer (Dr Sarah Mackenzie) who is acknowledged.

Major comments:

1) To clarify the clinical significance (not only statistically significant) of this study, authors should also explain the contexts in this platinum resistant disease field, specifically concerning the efficacy outcomes of the major clinical trials in the similar setting. This should include at least AURELIA, SORAYA, MIRASOL, CORT125134- 552 studies as stated in the Reviewers’ comments rebuttal (Major comments 1-2 of reviewer#3). It is essential to highlight the differences to understand the clinical impact of this trial compared to other reported studies given that most of them became the standard of care therapy options.

Author response: we have added a section in the discussion including these studies.

2) Figure 3:

(1) Please add a legend to specify in a different position that the symbol “P” refers to prior PARPi exposure. It can be misleading to include prior PARPi and PARPi naive without clarification in the best response legend.

Author response: the legend of Fig 3 has been updated.

(2) Please clarify why patients o continued the treatment beyond progression as explained in the rebuttal to reviewers#3 and #4.

Author response: we have added a note in the figure legend of the swimmer according to the explanation provided in the previous response to reviewers as to why patients stayed on therapy after progression (ie per protocol, it was left to the discretion of the investigator, to continue treatment in settings of clinical benefit. Note that, in most of those cases, progression occurred on one lymph node only, or one peritoneal lesion, in patients who previously responded to the therapy. In some of those patients, we have observed a further long-lasting stabilization of the disease.

(3) Please disclose the reason why the swimmer plot excluded three patients who were not evaluable for RECIST response.

Author response: the text in the figure legend of the swimmer has been modified to explicitly state the patients were excluded because they were not evaluable for response (progression was clinically symptomatic).

Minor comments

1. Safety section:

(1) Please specify the threshold of the “most common AEs” in the main text according to table 4 (20%).

Author response: modified as requested.

(2) Please edit the following sentence as suggested: withdrew from treatment due to an AE: including anemia (6 patients – olaparib)

Author response: modified as requested.

2. Discussion section:

The following sentence (page 9) “This may be due to differences in the extent of prior treatment (patients had not received PARP inhibitors and immunotherapy approximately half of our cohort had received prior PARP inhibitors, whereas in MEDIOLA prior PARP inhibitor exposure was not permitted). “ appears to be misinterpretation of the data because the author’s justification is for prior PARPi exposed group while the authors’ data of 47% ORR seen in the 15 patients are for PARPi naïve group and 77% ORR in MEDIOLA study which is also for PARPi naïve group.

Author response: sentence deleted

(4) Please add in the study limitations the use of archival tissues followed by an explanation about the reliability of the translational data.

Author response: text added to the discussion.

Reviewer #4 (Remarks to the Author):

I have read the responses and revisions. The authors addressed the comments adequately and the manuscript is improved. I had the following minor comments.

1. On page 5, please insert “/ missing status” in the text since this phase was added to Tables 2 and 3.

Author response: modified as requested.

2. In Supplementary Table 3, it would be better if row percentage is provided rather than the column percentage. So, the response could be compared between prior PARP inhibitor and no prior PARP inhibitor.

Author response: We thank the reviewer for this pertinent remark – we have updated the table, and the text accordingly.

3. Please provide the method that was used to calculate the 95% CI of the objective response rate in Tables 2 and 3.

Author response: Added to methods

4. In the new Figure 3, 'P' indicates prior PARP inhibitor therapy which is not a response category. Please remove 'P' from Best response according to RECIST.

Author response: Fig 3 has been updated

5. Please change Figure 3D, 3F in the text to Figure 4D, 4F.

Author response: thank you - text corrected

6. Thank you for providing the non-progression Kaplan-Meier curves of the two cohorts in the new Figure 2. It would help to understand the non-progression rates during the study. It appears that both cohorts had similar non-progression curves which might be due to the higher prior PARP inhibitor rate in the platinum sensitive cohort at baseline (52% vs 32%). In the platinum resistant cohort, the alternative hypothesis was that the non-progressive disease rate at 3 months was greater than 50% (historical control). For the platinum sensitive cohort, what was the rationale of choosing 65% non-progressive disease rate at 6 months as the null hypothesis? It was not stated neither in the protocol nor manuscript.

Author response: In the OCEAN trial (Aghajanian *et al*, 2012) with a test arm of gemcitabine / carboplatin / bevacizumab, the 12 month-PFS rate in patients with first platinum-sensitive relapse was 60%. Despite limited comparison with the BOLD population, we chose an « ambitious » threshold of 65% in a more heavily pretreated population.

Aghajanian C, Blank SV, Goff BA, Judson PL, Teneriello MG, Husain A, Sovak MA, Yi J, Nycum LR (2012) OCEANS: a randomized, double-blind, placebo-controlled phase III trial of chemotherapy with or without bevacizumab in patients with platinum-sensitive recurrent epithelial ovarian, primary peritoneal, or fallopian tube cancer. *J Clin Oncol* **30**: 2039–2045, doi:10.1200/JCO.2012.42.0505.

REVIEWERS' COMMENTS

Reviewer #3 (Remarks to the Author):

The reviewer appreciates the authors' response and a major revision of the manuscript. However, the reviewer is afraid that findings from the present study doesn't meet the high standard of readership of the Nature Communications given that this triplet therapy (olaparib [PARP inhibitor] + durvalumab [PD-L1 inhibitor] + bevacizumab [VEGF/VEGFR signalling inhibitor]) will unlikely add a significant novel insight into the ovarian cancer research field nor make a clinical impact to this population.

Specifically, from the recent findings from the single arm phase 2 multi-center study of the similar triplet combination (olaparib+pembrolizumab [PD1 inhibitor] + bevacizumab) (Nat Commun 2023, PMID: 37673858), Kim et al reported 6-months PFS rate of 88.6% in BRCAwt platinum sensitive recurrent ovarian cancer, with 22.9 months of the median follow up duration, meeting the pre-defined primary endpoint. They showed 22.4 months of the median PFS and 84% of 12-months PFS rate of 84.0% in platinum sensitive population.

Also, from the recent presentation of the randomized phase 2 multi-center study of 3-drug combination (olaparib + durvalumab [PD-L1 inhibitor] + cediranib [VEGFR TKI]) vs. standard of care chemotherapy in platinum resistant recurrent ovarian cancer patients with previous bevacizumab treatment (Lee et al. ESMO 2023), the study did not meet the primary endpoint and triplet therapy yielded the median PFS of 2.9 months in platinum resistant population.

That said, based on the findings from the current single arm study using the same concept of 3-drug combination (olaparib+durvalumab +bevacizumab), it is hard to make any conclusions considering the heterogenous population with mixed biological and clinical characteristics.

Major comment:

- 1) To clarify the clinical significance and relevance of the findings, the authors should provide the accurate contexts along with the latest information for the readers.

Reviewer #4 (Remarks to the Author):

There are two errors that needed to be corrected.

1. The authors responded that the 95% CI in Tables 2 and 3 were calculated using the exact binomial method. However, it appears that the CI was calculated using the normal approximation to the binomial distribution.
2. Please check the number of CR, PR, SD and PD in Figure 3. It seems that CR should be square sign, PR should be plus sign, SD should be circle sign and PD should be triangle sign.

RESPONSES TO REVIEWERS' COMMENTS

Reviewer #3 (Remarks to the Author):

The reviewer appreciates the authors' response and a major revision of the manuscript. However, the reviewer is afraid that findings from the present study doesn't meet the high standard of readership of the Nature Communications given that this triplet therapy (olaparib [PARP inhibitor] + durvalumab [PD-L1 inhibitor] + bevacizumab [VEGF/VEGFR signalling inhibitor]) will unlikely add a significant novel insight into the ovarian cancer research field nor make a clinical impact to this population.

Specifically, from the recent findings from the single arm phase 2 multi-center study of the similar triplet combination (olaparib+pembrolizumab [PD1 inhibitor] + bevacizumab) (Nat Commun 2023, PMID: 37673858), Kim et al reported 6-months PFS rate of 88.6% in BRCAwt platinum sensitive recurrent ovarian cancer, with 22.9 months of the median follow up duration, meeting the pre-defined primary endpoint. They showed 22.4 months of the median PFS and 84% of 12-months PFS rate of 84.0% in platinum sensitive population.

Also, from the recent presentation of the randomized phase 2 multi-center study of 3-drug combination (olaparib + durvalumab [PD-L1 inhibitor] + cediranib [VEGFR TKI]) vs. standard of care chemotherapy in platinum resistant recurrent ovarian cancer patients with previous bevacizumab treatment (Lee et al. ESMO 2023), the study did not meet the primary endpoint and triplet therapy yielded the median PFS of 2.9 months in platinum resistant population.

That said, based on the findings from the current single arm study using the same concept of 3-drug combination (olaparib+durvalumab +bevacizumab), it is hard to make any conclusions considering the heterogenous population with mixed biological and clinical characteristics.

Major comment:

1) To clarify the clinical significance and relevance of the findings, the authors should provide the accurate contexts along with the latest information for the readers.

Author response: We have added text and reference to the requested studies in the discussion to provide additional context.

Reviewer #4 (Remarks to the Author):

There are two errors that needed to be corrected.

1. The authors responded that the 95% CI in Tables 2 and 3 were calculated using the exact binomial method. However, it appears that the CI was calculated using the normal approximation to the binomial distribution.

Author response: We thank the reviewer for spotting this error and have updated the results of the 95% CI for Objective Response Rate estimations in Tables 2 and 3 and in the methodology section, with the exact method (Clopper-Pearson method) accordingly.

2. Please check the number of CR, PR, SD and PD in Figure 3. It seems that CR should be square sign, PR should be plus sign, SD should be circle sign and PD should be triangle sign.

Author response: We thank the reviewer for spotting this error. The figure legend has been updated accordingly.